

# Ozone air quality simulations with WRF-Chem (v3.5.1) over Europe: Model evaluation and chemical mechanism comparison

Kathleen A. Mar[1], Narendra Ojha[2], Andrea Pozzer[2], and Tim M. Butler[1]

[1]Institute for Advanced Sustainability Studies, Potsdam, Germany
[2]Atmospheric Chemistry Department, Max Planck Institute for Chemistry, Mainz, Germany

*Correspondence to:* K. A. Mar (Kathleen.Mar@iass-potsdam.de)

**Abstract.** We present an evaluation of the online regional model WRF-Chem over Europe with a fo-
cus on ground-level ozone ($O_3$) and nitrogen oxides ($NO_x$). The model performance is evaluated for
two chemical mechanisms, MOZART-4 and RADM2, for year-long simulations. Model-predicted
surface meteorological variables (e.g., temperature, wind speed and direction) compared well over-
all with surface-based observations, consistent with other WRF studies. WRF-Chem simulations
employing MOZART-4 as well as RADM2 chemistry were found to reproduce the observed spatial
variability in surface ozone over Europe. However, the absolute $O_3$ concentrations predicted by the
two chemical mechanisms were found to be quite different, with MOZART-4 predicting $O_3$ concen-
trations up to 20 $\mu g\,m^{-3}$ greater than RADM2 in summer. Compared to observations, MOZART-4
chemistry overpredicted $O_3$ concentrations for most of Europe in the summer and fall, with a sum-
mertime domain-wide mean bias of +10 $\mu g\,m^{-3}$ against observations from the AirBase network. In
contrast, RADM2 chemistry generally led to an underestimation of $O_3$ over the European domain in
all seasons. We found that the use of the MOZART-4 mechanism, evaluated here for the first time
for a European domain, led to lower absolute biases than RADM2 when compared to ground-based
observations. The two mechanisms show relatively similar behavior for $NO_x$, with both MOZART-4
and RADM2 resulting in a slight underestimation of $NO_x$ compared to surface observations. Further
investigation into the differences between the two mechanisms revealed that the net midday photo-
chemical production rate of $O_3$ in summer is higher for MOZART-4 than for RADM2 for most of
the domain. The largest differences in $O_3$ production can be seen over Germany, where net $O_3$ pro-
duction in MOZART-4 is seen to be higher than in RADM2 by 1.8 $ppb\,hr^{-1}$ (3.6 $\mu g\,m^{-3}\,hr^{-1}$)
or more. We also show that, while the two mechanisms exhibit similar $NO_x$-sensitivity, RADM2 is
approximately twice as sensitive to increases in anthropogenic VOC emissions as MOZART-4. Ad-



ditionally, we found that differences in reaction rate constants for inorganic gas phase chemistry in
MOZART-4 vs. RADM2 accounted for a difference of 8 µg m$^{-3}$ in O$_3$ predicted by the two mecha-
nisms, whereas differences in deposition and photolysis schemes explained smaller differences in in
O$_3$. Our results highlight the strong dependence of modeled surface O$_3$ over Europe on the choice
of gas phase chemical mechanism, which we discuss in the context of overall uncertainties in pre-
diction of ground-level O$_3$ and its associated health impacts (via the health-related metrics MDA8
and SOMO35).



## 1 Introduction

Tropospheric ozone ($O_3$) is an air pollutant with adverse effects on human and ecosystem health
as well as a short-lived climate forcer with a significant warming effect (e.g., Monks et al., 2015;
Stevenson et al., 2013; WHO, 2003). In Europe, ozone pollution remains a problem: the European
Environmental Agency reports that between 2010 and 2012, 98% of Europe's urban population was
exposed to $O_3$ levels in exceedance of the WHO air quality guideline (EEA, 2014), leading to more
than 6000 premature deaths annually (Lelieveld et al., 2015). This is despite the fact that European
emissions of ozone precursors, in particular nitrogen oxides ($NO_x$) and volatile organic compounds
(VOCs), have decreased significantly since 1990. The persistence of unhealthy levels of ozone in
Europe can be attributed to increases in hemispheric background ozone (Wilson et al., 2012) as well
as the non-linear relationship between $O_3$ and levels of precursor species $NO_x$ and VOC (EEA,

41 2014).

Air quality models are employed to understand the drivers of air pollution at a regional scale and to
evaluate the roles of and interactions between emissions, meteorology and chemistry. These models
fall into two broad categories: offline Chemistry-Transport Models (CTMs), in which meteorology is
calculated separately from model chemistry, and "online" models, the category to which WRF-Chem
belongs, in which the meteorology and chemistry are coupled, meaning they are solved together in
a physically consistent manner (e.g., Zhang, 2008). The meteorology and chemistry components in
WRF-Chem use the same horizontal and vertical grids and same timestep, eliminating the need for
temporal interpolation (e.g., Grell et al., 2004, 2005).
Air quality modeling studies over the European region have predominantly utilized CTMs, ex-
amples of which include EMEP (Simpson et al., 2012), CHIMERE (Terrenoire et al., 2015), and
LOTOS-EUROS (Schaap et al., 2008). Recently, application of WRF-Chem over Europe has in-
creased (e.g., Solazzo et al., 2012a, b; Tuccella et al., 2012; Zhang et al., 2013a, b; Baklanov et al.,
2014). However, only a limited number of studies dedicated to the evaluation of WRF-Chem simu-
lated meteorology and chemistry for the European domain are available in the literature. The study of
Tuccella et al. (2012) evaluated the performance of WRF-Chem using the RADM2 chemical mecha-
nism by comparing domain-wide average values against observations of meteorology and chemistry.
However, an evaluation of the spatial distribution of model-simulated meteorology and trace gases
is missing. This type of spatial information is extremely pertinent for air quality management appli-
cations, where model performance at a national scale can become more relevant than performance
metrics applied to the whole of Europe; this information gets lost when only comparing quantities
that have been averaged over the entire domain. Additionally, Tuccella et al. (2012) utilized time-
invariant chemical boundary conditions, which the authors suggested misrepresented the seasonal
changes in the intercontinental transport (Tuccella et al., 2012). In addition to the study of Tuccella
et al. (2012), Zhang et al. (2013b) evaluated the performance WRF-Chem-MADRID (Zhang et al.,
2010), an unofficial version of WRF-Chem coupled to the Model of Aerosol Dynamics, Reaction,



Ionization, and Dissolution (MADRID), over Europe for the month of July 2001, employing the gas-
phase mechanism CB05 (Yarwood et al., 2005). This detailed study provides a valuable reference
for comparison to the present work, but their simulations are only for one month, rather than the
complete seasonal cycle.
Several groups contributed WRF-Chem simulations to the AQMEII project (phase 1 and phase 2)
for the European domain (Solazzo et al., 2012b; Im et al., 2015). In AQMEII phase 1, two differ-
ent WRF-Chem simulations were part of the model ensemble for Europe, but evaluation of model
performance for ozone focused on evaluation of the ensemble (Solazzo et al., 2012b), rather than
on individual members. In fact, in the analysis of Solazzo et al. (2012b), individual models were
anonymized, meaning the performance statistics for the WRF-Chem ensemble members are not ex-
plicitly presented. The evaluation of model performance with respect to ozone in AQMEII phase 2
(Im et al., 2015) provides more information on the model performance of the contributing WRF-
Chem ensemble members for the European domain. In AQMEII phase 2, seven different WRF-Chem
runs were part of the ensemble. Of these seven simulations, four of them used the gas phase chemical
mechanism RADM2 (Stockwell et al., 1990), two used the mechanism CBMZ (Zaveri and Peters,
1999), and one used the mechanism RACM (Stockwell et al., 1997; Geiger et al., 2003). All WRF-
Chem simulations for Europe in AQMEII phase 2 tended to underestimate ozone concentrations,
with annual average normalized mean bias ranging from -1.6 to -15.8 %, depending on the ensemble
member.
The purpose of the present study is to perform a detailed evaluation of meteorology and gas phase
chemistry simulated by WRF-Chem, including the spatial and seasonal variations over a full year
seasonal cycle using time-varying chemical boundary conditions. This evaluation is performed for
two different gas phase chemical mechanisms within WRF-Chem, MOZART-4 (Emmons et al.,
2010) and RADM2 (Stockwell et al., 1990). As discussed above, the RADM2 mechanism has been
popularly used in WRF-Chem for simulation over Europe (Tuccella et al., 2012; Im et al., 2015). The
MOZART-4 chemical mechanism has been widely used with WRF-Chem for regional air quality ap-
plications outside of Europe (e.g., Pfister et al., 2013; Im et al., 2015). To the authors' knowledge,
however, WRF-Chem with MOZART-4 has never been applied and evaluated over a European do-
main.
The simultaneous evaluation of WRF-Chem with two different chemical mechanisms further al-
lows us to evaluate the sensitivity of $O_3$ and $NO_x$ to the choice of chemical mechanism in a setup
where the differences in model physics and other parameters are minimized. This is in contrast to
the study of Im et al. (2015), where the various WRF-Chem ensemble members also used different
schemes for model physics. Coates and Butler (2015) recently investigated the sensitivity of the pro-
duction of odd oxygen ($O_x$, a proxy for production of $O_3$) to the choice of chemical mechanism using
a box model, and found that choice of chemical mechanism led to differences in $O_3$ concentrations
on the order of 10 ppb under idealized conditions, although differences between the MOZART-4



and RADM2 chemical mechanisms tended to be closer to 5 ppb. In another box model study, Knote
et al. (2015) investigated the sensitivity of $O_3$, $NO_x$, and other radicals to the different gas-phase
chemical mechanisms used in the models that contributed to the AQMEII phase-2 intercomparison
project. Knote et al. (2015) found that the choice of chemical mechanism is responsible for a 5%
uncertainty in predicted $O_3$ concentrations and a 25% uncertainty in predicted $NO_x$ concentrations.
The present study builds on the work of Coates and Butler (2015) and Knote et al. (2015) by
comparing two chemical mechanisms within an online coupled regional air quality model. The use
of WRF-Chem provides an advantage in that it is compatible with multiple different chemical mech-
anisms, allowing us to test the effect of different chemistry with minimal confounding factors due to
differences in model physics, etc. Furthermore, the use of an online regional model rather than a box
model allows us to examine the sensitivity of model-predicted concentrations to the choice of chem-
ical mechanism under more realistic conditions, in which variations in meteorology and dynamics is
fully included. Parameters such as radiation are allowed to vary realistically, and different chemical
regimes ($NO_x$- vs. VOC-limited) are present (e.g., in different seasons and in different parts of the
model domain).
Chemical mechanism comparisons have also been undertaken previously using 3-D regional air
quality models, though the majority have focused on comparing the SAPRC-99 mechanism (Carter,
1990) with versions of the Carbon Bond mechanism (Gery et al., 1989) over a U.S. domain (Luecken
et al., 2008; Faraji et al., 2008; Yarwood et al., 2003; Zhang et al., 2012). Two additional studies have
compared versions of the RACM mechanism with RADM2 (Mallet and Sportisse, 2006) and CB05
(Kim et al., 2010) using the model Polyphemus (Mallet et al., 2007) for a European domain. Typ-
ically, these studies found that simulations using two different chemical mechanisms led to differ-
ences in $O_3$ on the order of 5-10 ppb (Luecken et al., 2008; Zhang et al., 2012; Mallet and Sportisse,
2006; Kim et al., 2010), although extreme differences of 30-40 ppb were observed between SAPRC-
99 and CB-IV mechanisms when simulating high ozone episodes (Faraji et al., 2008; Yarwood et al.,

129 2003).

In this paper, the model configuration, including emissions and initial and boundary conditions, is
described in Section 2. A description of observational datasets for meteorology and chemistry and
the evaluation methodology is provided in Section 3. Results for the model evaluation and intercom-
parison of two chemical are presented in Section 4 followed by a summary and concluding remarks
in Section 5.



## 2 Model Description and Setup

### 2.1 WRF-Chem

This study utilizes the Weather Research and Forecasting with Chemistry (WRF-Chem) model (http://ruc.noaa.gov/wrf/WG11) version 3.5.1. WRF-Chem has been developed collaboratively by NOAA, DOE/PNNL, NCAR and other research institutes (https://www2.acd.ucar.edu/wrf-chem).

We defined our simulation domain on the Lambert projection. The model domain is centered at $15°$ E, $52°$ N, and covers nearly the entire European region. The horizontal resolution is chosen to be $45\,km \times 45\,km$. The model domain has 115 and 100 grid points in the west-east and south-north directions respectively.

We have used 35 vertical levels in the model starting from surface to $10\,hPa$. The lowest model level corresponds to an approximate altitude of $50\,m$ above the surface. Tests have shown that surface layer concentrations in this configuration are effectively the same as when the lowest model level is at a height of $14\,m$, but with no urban surface physics scheme (the urban physics scheme is incompatible with a 14-m model level). Geographical data including terrain height, soil properties, albedo, etc. are interpolated primarily from USGS (United States Geological Survey data (Wang et al., 2014)) at $30\,sec$ resolution. The land use classification has been interpolated from the CORINE data (EEA, 2012) at $250\,m$ resolution, which was then mapped to the USGS land use classes used by WRF (see Kuik et al., 2016).

Model simulations are conducted for the period of 23 December 2006 to 31 December 2007. The first week of output was treated as model spin up and has been discarded. The instantaneous model output, stored every hour, has been used for the analysis. The different options used in this study to parametrize the atmospheric processes are listed in Table 1. A namelist is available in the Supplementary Material.

The initial and lateral boundary conditions for the meteorological fields were provided from the ERA-interim reanalysis dataset available from ECMWF (http://www.ecmwf.int/en/research/climate-reanalysis/era-interim). This data is available every $6\,hours$ with a spatial resolution of approximately $80\,km$ (T255 spectral). In order to limit the errors in the WRF simulated meteorology the Four Dimensional Data Assimilation (FDDA) has been applied. In the FDDA, temperature is nudged at all the vertical levels with a nudging coefficient of 0.0003. The horizontal winds are nudged at all the vertical levels, except within the PBL, with the nudging coefficient of 0.0003. Sensitivity studies performed showed that nudging of water vapor highly suppressed the precipitation over Europe in a manner inconsistent with observations. As such, water vapor is not nudged in our simulations. This also follows the approach of, e.g., Miguez-Macho et al. (2004) and Stegehuis et al. (2014). The nudging coefficients for temperature and winds have been chosen following previous studies (Stauffer et al., 1991; Liu et al., 2012). The time step for the simulations has been set at $180\,s$.



Initial and boundary conditions for chemical fields in WRF-Chem are used from the MOZART-
4/GEOS5 simulations (http://www.acd.ucar.edu/wrf-chem/mozart.shtml), with a horizontal resolu-
tion of $1.9° \times 2.5°$ and 56 pressure levels. MOZART-4/GEOS-5 simulations use meteorology from
the NASA GMAO GEOS-5 model and emissions based on ARCTAS inventory (http://www.cgrer.
uiowa.edu/arctas/emission.html).

### 2.2  Emissions

Anthropogenic emissions of CO, $NO_x$, $SO_2$, NMVOCs, $PM_{10}$, $PM_{25}$, and $NH_3$ are used from the
TNO-MACC II emission inventory for Europe (Kuenen et al., 2014), for the year 2007. These emis-
sions are provided as yearly totals by source sector on a high-resolution ($7\,km \times 7\,km$) grid. The
TNO-MACC II emission inventory is based on emissions reported by member countries to the Eu-
ropean Monitoring and Evaluation Program (EMEP), which are then further refined to fill gaps and
correct errors and obvious inconsistencies. Emissions are temporally disaggregated based on sea-
sonal, weekly and diurnal cycles provided by Denier van der Gon et al. (2011); Schaap et al. (2005).
These temporal profiles vary by source sector according to the SNAP (Selected Nomenclature for
Sources of Air Pollution) convention. NMVOC emissions are split into modeled NMVOC species
(e.g., ethane, aldehydes) based on von Schneidemesser et al. (2016). $NO_x$ is emitted as 90% NO and
10% $NO_2$ by mole. Emissions are distributed into the first seven model vertical layers (the surface
and the first 6 model layers above the surface) based on sectoral averages from (Bieser et al., 2011),
although model runs showed little sensitivity to the distribution of emissions above the surface layer.
The model domain used in this study is larger than the European domain used in the TNO-
MACC II inventory (Kuenen et al., 2014). Emissions at our domain edges were filled using the
Hemispheric Transport of Air Pollution (HTAP v2.2) emission inventory for the year 2008 (http:
//edgar.jrc.ec.europa.eu/htap_v2/index.php). The HTAP v2 data, described in detail by Janssens-
Maenhout et al. (2015), is harmonized at a spatial resolution of 0.1° x 0.1° and available with
monthly time resolution. In our model simulations, no additional weekly or diurnal profiles were
applied to the HTAP v2 emissions. Furthermore, all emissions from HTAP were emitted into the
surface model layer. Because HTAP emissions were only used at the grid "edge," the differences
in temporal and vertical resolution of emissions used for HTAP is not expected to have a signifi-
cant impact on model results. An example of emissions processed for model input is shown in the
Supplementary Material.
Biomass burning emissions are from the Fire Inventory from NCAR (FINN), Version 1 (Wiedin-
myer et al., 2011). To avoid the double counting of emissions from agricultural burning (i.e., assum-
ing that the FINN product captures large-scale agricultural burning), emissions of the combustion
species CO, $NO_x$, and $SO_2$ from SNAP category 10 (Agriculture) in the TNO-MACC II inventory
were not included in model simulations, at the suggestion of H.A.C. van der Gon (personal commu-



nication, 2015). Biogenic Emissions are calculated online based on weather and land use data using
the Model of Emissions of Gases and Aerosols from Nature (MEGAN) (Guenther et al., 2006).

## 2.3 Model Chemistry

The two year-long WRF-Chem simulations performed for this study are summarized in Table 2.
In the MOZART simulation, gas phase chemistry is represented by the Model for Ozone and and
Related chemical Tracers, version 4 (MOZART-4) mechanism (Emmons et al., 2010). Tropospheric
chemistry is represented by 81 chemical species, which participate in 38 photolysis and 159 gas-
phase reactions. The MOZART-4 mechanism includes explicit representation of the NMVOCs ethane,
propane, ethene, propene, methanol, isoprene, and $\alpha$-pinene. Other NMVOC species are represented
by lumped species based on the reactive functional groups. In the WRFV3.5.1 code, two bug fixes
have been included for the MOZART-4 mechanism: the $NH_3$+OH rate constant has been corrected
following Knote et al. (2015), and a correction has been made to treatment of the vertical mixing
of MOZART-4 species (A.K. Peterson, personal communication). In the WRF-Chem simulations,
we use the version of MOZART-4 coupled to the simple GOCART aerosols mechanism (Acker-
mann et al., 1998b), known as the MOZCART mechanism. In this paper, we limit our analysis to
gas-phase species. Because of this focus, and to simplify the interpretation the mechanism intercom-
parison (see below), all aerosol radiative feedbacks (i.e., both direct and indirect effects) are turned
off in all model simulations in this study.
In the RADM2 simulation, gas phase chemistry is represented by the second generation Regional
Acid deposition Model (RADM2) (Stockwell et al., 1990). This mechanism has 63 chemical species
which participate in 21 photolysis and 136 gas phase reactions. The NMVOC oxidation in RADM2 is
treated in a less-explicit fashion than in MOZART, in which ethane, ethene and isoprene are the only
species treated explicitly and all other NMVOCs are assigned to lumped species based on OH reac-
tivity and molecular weight. In WRF-Chem, RADM2 is coupled to the MADE/SORGAM aerosol
module, which is based on the Modal Aerosol Dynamics Model for Europe (MADE) (Binkowski
and Shankar, 1995; Ackermann et al., 1998a) and Secondary Organic Aerosol Model (SORGAM)
(Schell et al., 2001) . However, as noted above, in this study we focus our analysis on gas-phase
chemistry.
In both the RADM2 and MOZART simulations, the chemical mechanism code was generated
with the Kinetic Pre-Processor (KPP) (Damian et al., 2002; Sandu and Sander, 2006), and equations
are solved using a Rosenbrock-type solver. Note that when using RADM2 chemistry, there are two
different solvers available within WRF-Chem. We chose to use the KPP chemistry and Rosenbrock
solver to be consistent with the MOZART runs, and also because the alternative QSSA chemistry
solver has been shown to have problems representing $NO_x$ titration (Forkel et al., 2015). In partic-
ular, the QSSA treatment of RADM2 chemistry was found to result in an under-representation fo
nocturnal ozone titration for areas with high NO emissions.



## 3 Observational datasets

A summary of the observational datasets used for model evaluation can be found in Table 3.

### 3.1 Meteorology

Since WRF-Chem couples the meteorology simulations online with the chemistry, we begin by evaluating the modeled meteorological fields against observations which are driving the simulations of chemical fields. In this study, the WRF-Chem simulated meteorological fields are evaluated against the in situ measurements of mean sea level pressure (MSLP), 2-meter temperature (T2) and 10-meter wind speed and direction (WS10 and WD10, respectively) from the Global Weather Observation dataset provided by the British Atmospheric Data Center (BADC). We chose these meteorological variables for the evaluation as these are expected to have the most significant influence on the gas-phase chemistry, which is the main focus of this study.

### 3.2 Chemistry

#### 3.2.1 EMEP Network

The EMEP observational dataset provides surface measurements of pollutant concentrations, including tropospheric ozone and its precursors, at stations chosen to be representative of regional background pollution (see, e.g., Tørseth et al., 2012). The regional focus is in keeping with the goals of the Convention on Long-range Transboundary Air Pollution (CLRTAP), under which this network is administrated.

#### 3.2.2 AirBase Network

AirBase is the public air quality database of the European Environmental Agency (EEA), and represents a much denser network of monitoring than the EMEP network (http://www.eea.europa.eu/data-and-maps/data/airbase-the-european-air-quality-database-7). Because of the relatively coarse horizontal resolution in this model study, model output is only compared against AirBase stations that are classified as "rural background." Some AirBase stations are also part of the EMEP network; the subset of AirBase stations used in this study exclude any stations that are also part of the EMEP network (since they are already included in the evaluation against EMEP observations).

### 3.3 Evaluation methodology

Stations were excluded from our season-by-season analysis if the temporal coverage was less than 75%, i.e., if missing or flagged hourly (or 3-hourly) data represented more than 25% of the hourly (or 3-hourly) time series over the entire season. For sensitivity studies that consider the month of July only, stations were considered that had at least 75% temporal coverage for the month. This



criteria was applied for all meteorological and chemistry observations. For comparison of model
output to in situ observations, the model gridcell that is closest to the latitude, longitude location
of the measurement station was chosen. Statistics calculated include the mean, mean bias (MB),
normalized mean bias (NMB), mean fractional bias (MFB) and the temporal correlation coefficient
(r). The domain-wide statistics presented in Tables 4 - 9 were calculating by first calculating the
statistical quantity hour-by-hour at each station, and then averaging these values over all times (in
the season) and all stations. Definitions of calculated statistical quantities can be found in Appendix
B.
From hourly concentrations of $O_3$, either observed or modeled, additional ozone metrics for health
impacts are calculated. MDA8 is defined as the maximum daily 8-hour mean ozone, in accordance
with the European Union's Air Quality Directive. Note that, for calculation of MDA8, a missing
value was assigned if one or more hours of data in the 8-hour average were missing. SOMO35 is de-
fined as the sum of MDA8 levels over 35 ppb ($70\,\mu g\,m^{-3}$) over a year, in units of concentration·days,
following Regional Office for Europe (2008).
$$SOMO35 = \frac{365}{N_{valid}} \sum_{iday} max(0, C_{iday} - 70\,\mu g\,m^{-3})$$
where $N_{valid}$ is the number of valid (i.e., non-missing) daily values.

## 288   4   Results and Discussion

### 289   4.1   Evaluation of Meteorology

Table 4 shows a summary of domain-wide statistics evaluating the model simulations against obser-
vations of meteorological variables MSLP, T2, WS10 and WD10; the spatial distribution of these
statistics shown in Figures 1-3 for temperature and wind variables. (Shown in Table 4 and Fig-
ures 1-3 are model output from the MOZART simulation; differences between the MOZART and
RADM2 simulations are negligible, as expected based on fact that simulations were run without
aerosol-radiative feedbacks.) MSLP has been reproduced over the entire European domain with a
high degree of skill in every season, with negligible bias (domain-averaged NMB and MFB are zero
in all seasons) and temporal correlation coefficients (r values) of 0.98 or greater (see also Figure S2
in the Supplementary Material).
The spatial distribution of seasonal average T2 in the model and observations is shown in Figure 1,
along with the spatial variation in mean bias and temporal (3-hourly) correlation. Overall the spatial
variability in measured T2 is found to be well-reproduced by WRF-Chem during all the seasons.
The absolute values of mean biases in T2 were generally found to be lower than $1°\,C$. Larger biases
in T2 can be found in the Alps, in particular during winter, where T2 is often overpredicted by
more than $1°\,C$ (Figure 1). This larger bias over mountainous regions, also found in a previous
study (Zhang et al., 2013a), is likely due to the complex mountain terrain and associated unresolved



local dynamics. The r values are generally found to be more than 0.9 in all the seasons and show no significant geographical variation, indicating that the model is able to reproduce the hourly variations in near surface temperature. Averaged over the entire domain, the mean bias in T2 varies from -0.4 to + 0.3° C depending on the season (Table 4). varies from -0.4 to + 0.3° C depending on the season (Table 4).

The spatial variability in wind speeds, including the seasonality with strongest winds during the winter have been reproduced by the model (Figure 2). However, the model tends to overestimate winds speeds with larger biases (2 m/s or more) during the winter and fall. The regions showing greater bias in wind speed include the Alps, coastal regions, and the low-lying areas of northern Germany and Denmark (Figure 2). The temporal correlation of wind speed is generally above 0.7 in the northern half of the domain, but is lower (0.4-0.6) in the southern part of the domain, in areas in the Alps and close to the Mediterranean (Figure 2). Similar behavior for modeled wind speed is reported by Zhang et al. (2013a), who attributes the overestimation in wind speeds primarily to poor representation of surface drag exerted by unresolved topographical features, which results in model limitations in simulating circulation systems such as sea breeze and bay breeze. An overview of the statistics for wind direction is presented in Table 4, with the spatial distribution shown in Figure 3. Wind comes dominantly from the west and south, and the mean bias in wind direction is between 20 and 30 degrees depending on the season. Similar to the patterns seen for wind speed, areas with complex topography (the Alps, the Balkans, the Mediterranean coast) show the largest biases and the lowest correlations for wind direction.

Overall, we find that WRF-Chem is capable of reproducing the spatial and temporal variations in the European meteorological conditions reasonably well, in a manner consistent with previous studies (e.g. Zhang et al., 2013a).

## 4.2 Evaluation of Chemistry

### 4.2.1 Ozone

We begin the evaluation of chemistry by examining the seasonal average surface $O_3$ distribution over Europe from the MOZART simulation, as shown in Figure 4. Predicted surface $O_3$ distributions show a clear seasonality, with maximum concentrations during summer. In all seasons, surface $O_3$ concentrations are highest over the Mediterranean region, with values during the spring and summer greater than 110 $\mu g \, m^{-3}$. Simulated concentrations reproduce the north-south gradient in $O_3$ seen in the ground-based observations. Figure 5 provides another comparison of seasonal average $O_3$ distributions in the model vs. the observations (from both the AirBase and EMEP networks) and additionally shows the spatial distribution of MB and r, the temporal (hourly) correlation coefficient; performance statistics are shown in Table 5 (against observations from the AirBase network) and Table 6 (against observations from the EMEP network). MOZART overpredicts $O_3$ concentrations



for most of Europe in the summer and fall. In winter and spring, MOZART tends to underestimate $O_3$
in north-central Europe, but overestimate $O_3$ in southern Europe. Hourly correlation coefficents for
$O_3$ are highest (greater than 0.6) in northern Europe (especially France, Germany, and the Benelux
region) and in Spain, but are lower (with values of approximately 0.4) throughout Italy and the
mountainous regions of the Alps. Notably, Italy and the Alps are the regions within our domain
that exhibit the highest biases and lowest correlations with respect to wind direction and speed
(Section 4.1), which could explain the poorer temporal correlation for $O_3$ in these areas.
Looking at Tables 5 and 6, we see some differences in the statistical performance of the MOZART
simulation when compared to the EMEP vs. the AirBase observational datasets. Considering the
EMEP observations over the whole domain (Table 6), MOZART slightly overpredicts $O_3$ in sum-
mer, with a summertime mean bias of $4\,\mu g\,m^{-3}$, whereas the summertime mean bias when compared
the AirBase network is $10\,\mu g\,m^{-3}$ (Table 5). In winter and spring, the bias (MB, NMB, and MFB)
in MOZART-predicted $O_3$ is more negative when compared to EMEP observations than to AirBase
observations. In fall, the sign of the domain-average bias changes if considering the model perfor-
mance against EMEP vs. AirBase observations. These differences likely reflect differences in the
character of the two observational networks. First, we expect that the Airbase rural background sites
considered here may be, on average, more influenced by local pollution sources than the EMEP sites,
which are selected to be representative of more remote regional background. Secondly, the geograph-
ical coverage of AirBase vs. EMEP sites for $O_3$ is slightly different (Supplementary Material). In
particular, coverage of the U.K. and the Nordic countries is almost exclusively via the EMEP net-
work, potentially giving the EMEP observations a northern bias in comparison to the AirBase-only
sites. Both features of the measurement networks could explain the lower values of the domain-wide
average $O_3$ observed at the EMEP vs. the AirBase stations.
In addition to evaluating the model's ability to simulate hourly $O_3$ concentrations, we also con-
sider MDA8 and SOMO35, two metrics designed to evaluate the impact of ozone on health. The
distribution of seasonal average values of MDA8 is shown in Figure 6 for the MOZART simulation.
The European Union's Air Quality Directive states that, as a long-term objective, MDA8 should not
exceed the threshhold value of $120\,\mu g\,m^{-3}$; as a target value this long-term objective should not be
exceeded on more than 25 days per year, averaged over 3 years. Figure 6 shows that, at some stations
in the Alps and in southern Italy during summer, the average value of MDA8 exceeds $120\,\mu g\,m^{-3}$.
As seen in Figure 7, the number of days when MDA8 exceeds the $120\,\mu g\,m^{-3}$ is greater than 25 in
spring alone for much of southern Europe, which is also captured well by the MOZART simulation.
MOZART tends to overpredict MDA8 and the days in exceedance of the target value in summer and
fall, consistent with the overestimation of hourly average $O_3$ during this season. Since the metric
MDA8 is, in effect, a measure of daytime ozone, it is always higher than the straight average of
hourly concentrations. As a consequence, MOZART shows greater bias in MDA8 than in average
$O_3$ in seasons where average $O_3$ is already overpredicted (Tables 5 and 6). In general, regional and



seasonal patterns for MDA8 simulated by MOZART are similar to those for simulated average $O_3$.
SOMO35, an indicator for cumulative annual exposure, is shown in Figure 8 for the year 2007.
MOZART is able to reproduce the north-south gradient of SOMO35 seen in the observations quite
well, while overpredicting the magnitude of SOMO35 by $2\,\mathrm{mg\,m^{-3}\cdot days}$ (Table 7).
WRF-Chem simulations using the RADM2 chemical mechanism show a spatial and seasonal
distribution of surface $O_3$ over Europe (Figures 9 and 10) that is qualitatively similar to that for
MOZART. The correlation coefficients for the MOZART and RADM2 simulations are also similar
in both magnitude in distribution. However, it is striking to note that the surface $O_3$ concentrations
predicted by two different chemical mechanisms are quite different, with RADM2 predicting aver-
age surface $O_3$ values that are approximately $20\,\mu\mathrm{g\,m^{-3}}$ lower than those predicted by MOZART
in spring and summer (c.f. Figures 4 and  9, Tables 5 and  8, and Tables 6 and  9). In contrast to
MOZART, RADM2 underpredicts $O_3$ throughout most of Europe in all seasons. An exception to
this is in southern Europe in winter, where RADM2, like MOZART, shows some overprediction of
$O_3$ concentrations in southern Europe, particularly near the Mediterranean. RADM2 also overpre-
dicts $O_3$ near the Mediterranean in fall (a season where MOZART overpredicts $O_3$ Europe-wide).
The general underprediction of $O_3$ concentrations in RADM2 means that the health metrics MDA8
and SOMO35 are also underpredicted (Tables 7- 8 and Figure 8). Overall, absolute biases (i.e., the
absolute value of MB, NMB, and MFB) are smaller for MOZART than for RADM2, indicating that
MOZART is more successful overall in reproducing European ground-level $O_3$.
Model biases for $O_3$ in both the MOZART and RADM2 simulations are in line with biases found
in other regional modeling studies for Europe. For instance, values for the NMB in European sum-
mertime $O_3$ ranged from less than -20% to greater than +20% depending on the ensemble member
in AQMEII (Solazzo et al., 2012b; Im et al., 2015), compared to values of -18% and +14% for the
RADM2 and MOZART simulations, respectively, in the present study. Zhang et al. (2013b) found
domain-wide values for NMB for $O_3$ ranging from +4.2% to +19.1% for the month of July 2001,
depending on their model configuration. Tuccella et al. (2012) report a domain-average mean bias
in $O_3$ of -1.4 $\mu\mathrm{g\,m^{-3}}$ averaged over the whole year. Although the work of Tuccella et al. (2012)
uses the RADM2 chemical mechanism and simulates the year 2007, similar to the RADM2 simula-
tion in the present study, there are several differences in model configuruation that could explain the
observed differences in predicted $O_3$, including the use of time-invariant chemical boundary condi-
tions, the use of the QSSA rather than the Rosenbrock chemical solver (which has been shown to
make a difference (see Forkel et al., 2015)), and the use of an alternate emissions inventory (from
EMEP).
The temporal correlation with hourly measurements for $O_3$ in this study are also in line with
other regional modeling studies of $O_3$ for Europe. Simulations with both chemical mechanisms lead
to reasonable correlations between the model-predicted and observed $O_3$ concentrations over the
entire domain, with r values generally in the range of 0.6-0.8 (Figures 5 and 10, Tables 5 and 8).



This is consistent with the hourly correlation coefficient for $O_3$ of 0.62 reported by Tuccella et al.
(2012), where their r value represents an average over the entire year of 2007. Zhang et al. (2013b)
also report correlation coefficients of 0.6-0.7 for hourly $O_3$ over the European domain (horizontal
resolution 0.5°) using the CB05 gas-phase chemical mechanism in WRF-Chem.

In addition to evaluating the performance of the MOZART and RADM2 simulations on their abil-

ity to reproduce ground-level ozone concentrations, we compare the observed sensitivity of modeled
$O_3$ to the choice of chemical mechanism to other studies that have investigated the uncertainty in
3-D model predictions associated with the choice of chemical mechanism. Knote et al. (2015) used
box model simulations based on AQMEII phase 2, and concluded that the uncertainty in predicted
$O_3$ in a 3-D model solely due to the choice of gas phase chemical mechanism should be of the order
of 5%, or 4 ppbv (8 $\mu g\,m^{-3}$). This is quite a bit smaller than the sensitivity to chemical mechanism
found in this study, where we see differences in summertime average $O_3$ of 20 $\mu g\,m^{-3}$, correspond-
ing to a relative difference of approximately 40%. Coates et al. (2016) have shown that accounting
for stagnant conditions in a box model increased the variability in predicted $O_3$ with temperature in a
way that better reproduced the variability seen in observational datasets and 3-D model simulations;
adding representation of stagnant conditions (which were not represented in Knote et al. (2015)) to
the box model also increased the sensitivity of predicted $O_3$ to the chemical mechanism. This re-
sult suggests that day-to-day variability in meteorological conditions and transport can enhance the
sensitivity of $O_3$ to chemical mechanism compared to what is seen in box models.

Another interesting basis for comparison is the study of Mallet and Sportisse (2006), who investi-

gate uncertainty in the CTM Polyphemus due to various physical parameterizations, including chem-
ical mechanism (comparing RACM and RADM2), using an ensemble approach. They estimated an
overall uncertainty in $O_3$ concentrations of 17% based on choices for physical parameterizations in
general, but identifed the choice of chemical mechanism along with the turbulent closure parame-
terization as the two most important drivers of this uncertainty. Simulations using the RACM vs.
RADM2 mechanisms yielded differences in average $O_3$ concentrations of 7-13 $\mu g\,m^{-3}$, depending
on the other parameterizations used. It is clear that the sensitivity of $O_3$ to the use of the MOZART
vs. RADM2 chemical mechanism in this study is large compared to other studies of mechanism
comparisons in 3-D models (see also  Luecken et al., 2008; Kim et al., 2010)), though even larger
absolute differences in hourly $O_3$ concentrations (up to 40 ppb, or 80 $\mu g\,m^{-3}$) have been found in
studies of episodic ozone (Faraji et al., 2008; Yarwood et al., 2003). It is possible that MOZART
and RADM2 as implemented in this study are examples of chemical mechanisms that are extremely
different from one another on a spectrum of other commonly-used mechanisms; the differences be-
tween the two mechanisms will be further explored in Section 4.3.



### 4.2.2 Nitrogen oxides


Seasonal average surface-level $NO_x$ for the MOZART simulation are shown in Figure 11. Several
hotspots in the spatial distribution of $NO_x$ mixing ratios are apparent, as expected based on the
intensity of emissions in these areas. $NO_x$ hotspots with concentrations of more than 30 $\mu g\,m^{-3}$
are visible over parts of France, Belgium, Germany and Russia. Similar high concentrations are
also seen over the marine regions close to Barcelona, Monaco, and southern France. As shown
in Table 5, the MOZART simulation slightly underpredicts domain-average $NO_x$ concentrations
for all seasons when comparing to AirBase observations. In Figures 12 and 13 we examine the
spatial distribution of $NO_x$ broken down into its components, $NO_2$ and NO, together with the spatial
distribution of MB and r. The MOZART simulation overestimates $NO_2$ in the U.K., northern France,
Belgium, and central Germany, all of which are regions known for having high $NO_x$ emissions and
concentrations. However this does not hold true for the Netherlands, a neighboring region with high
emissions where MOZART tends to underpredict rather than overpredict $NO_2$ concentrations. NO,
on the other hand, is significantly underpredicted compared to surface measurements throughout
the domain. This may be partially due to the relatively coarse horizontal resolution of the model, in
which fresh NO emissions are immediately diluted over a large area, and could also be a consequence
of model deficiencies in representing $NO_x$ chemical cycles. Artifacts related to reporting of low
NO concentrations approaching measurement detection limits could also play a role (observed time
series for NO typically show a baseline of 1-2 $\mu g\,m^{-3}$, whereas modeled concentrations reach a
baseline of zero).
Domain average temporal correlation coefficients (r) against hourly measurements of $NO_x$, $NO_2$,
and NO (Tables 5 and 6) range from approximately 0.2 to 0.5, which is lower than correlations for
$O_3$ but consistent with other studies, dicussed further below. In all seasons, the domain-averaged
temporal correlation coeffiecient is higher when compared to EMEP vs. AirBase observations. This
is attributed to lesser local influences and therefore better regional representativeness of the EMEP
stations. No exceptional patterns are seen in the spatial distribution of r for $NO_2$ or NO, although
correlation appears slightly better in the northern part of the domain. The MOZART simulation
shows the highest domain-average correlation coefficients (r) for $NO_x$, $NO_2$, and NO in winter and
fall, and the lowest domain-average r values in summer.
$NO_x$ predicted by the RADM2 simulation shows fairly similar behavior to $NO_x$ predicted by the
MOZART simulation (cf. Figures 12 and 14 and additional figures in the Supplementary Mate-
rial). In general, simulated $NO_x$ concentrations are slightly higher for MOZART than for RADM2.
Domain-wide average $NO_x$ concentrations predicted by MOZART are approximately 2 $\mu g\,m^{-3}$
higher than for RADM2 in all seasons except winter, where the difference is approximately 3 $\mu g\,m^{-3}$
(cf. Tables 5 and 8). The spatial distribution of MB for $NO_2$ for the RADM2 simulation generally
shows the same patterns as observed for the MOZART simulation, namely a slight overestimation
in the U.K., northern France, Belgium, and central Germany. Temporal correlation is also found to





show similar behavior to the MOZART simulation. An exception to the similarity observed between
the mechanisms for $NO_x$ can be seen over central Germany in winter, where MB values for $NO_2$ are
6-10 $\mu g\,m^{-3}$ for MOZART, but in the range of 0-6 $\mu g\,m^{-3}$ for RADM2 (ref supplementary mate-
rial for RADM2 plot). Differences in $NO_x$ concentrations predicted by the MOZART vs. RADM2
simulations are generally less than 20%, consistent with Knote et al. (2015), who conclude that un-
certainty due to choice in chemical mechanism leads to an uncertainty of up to 25% in 3-D model
simulations.
Performance of the present simulations with respect to $NO_2$ can also be compared to previous
published studies (note that none of the above-cited studies perform a validation for NO or $NO_x$).
Zhang et al. (2013b) reports NMB values of approximately -15% for $NO_2$ for WRF-Chem simu-
lations against hourly AirBase measurements for July 2001, in line with values of -12% and -19%
for the MOZART and RADM2 simulations in this study, respectively. Tuccella et al. (2012) report a
MB for $NO_2$ of -0.9 $\mu g\,m^{-3}$ averaged over the whole year; for comparison the RADM2 simulation
in this study shows a MB in the range of -2.5 to -1 $\mu g\,m^{-3}$ for fall, spring and summer, but a MB of
+0.67 $\mu g\,m^{-3}$ in summer. Evaluation of $NO_2$ was not treated in detail in the AQMEII studies, but
Im et al. (2015) report that the models for the European domain underestimate $NO_2$ by 9% to 45%.

### 4.3   Characterization of MOZART vs. RADM2 differences

In this section, we explore the differences in surface $O_3$ between the MOZART and RADM2 simula-
tions by examining net $O_3$, $NO_2$, and NO production rates as well as the $NO_x$- and VOC-sensitivity
of the two mechanisms. We further conducted sensitivity simulations to investigate the relative con-
tributions of different sources to the observed differences in surface $O_3$ predicted by MOZART and
RADM2. The month of July was chosen for the sensitivity simulations since $O_3$ concentrations over
Europe are highest during summer, and thus summer is most the most important season when con-
sidering air quality exceedances and health impacts of $O_3$. Additionally, MOZART and RADM2
show the largest differences in predicted $O_3$ during this season (see Tables 5 and 8).
To gain insight into model behavior for $O_3$, we added terms to the model output representing
hourly accumulated tendencies, i.e., the change in concentration of a species due to photochemistry
only, for July simulations using MOZART and RADM2. The hourly net photochemical production
rate was calculated as the difference in the accumulated tendency from one timestep to another. Fig-
ure 15 shows the average of the midday (11:00-14:00 CEST, or 9:00-12:00 UTC) photochemical
production rate of $O_3$ and $NO_x$ components for both the MOZART and RADM2 simulations. (Note
that the net photochemical production rate is shown here in $ppb\,hr^{-1}$ for more intuitive comparison
of production and loss of the different species on a mole basis; $\mu g\,m^{-3}$ was used in Section 4.2 be-
cause this is the unit in which limit and target values in the EU Air Quality Directive are expressed.)
Overall, the spatial variability as well as the magnitudes of net $O_3$ production rates are found to
be similar for MOZART-4 and RADM2 chemistry (Figure 15). For both mechanisms, the greatest



midday net $O_3$ production rates are found in southern Europe, particularly over the Mediterranean
and Atlantic coasts. The difference in net $O_3$ production rate between the two mechanisms is also
shown in Figure 15. MOZART exhibits greater net $O_3$ photochemical production rates than RADM2
for most of Europe, with the exception of the southeast corner of the domain (Greece, Turkey, and
the nearby Mediterranean), where net $O_3$ production rates are greater for RADM2. The difference
in net $O_3$ production rate (MOZART-RADM2) shows a large maximum over central Europe, cen-
tering over Germany and extending west and east into France and Poland. Over Germany, net $O_3$
production in MOZART is seen to be higher than in RADM2 by 1.8 $\mathrm{ppb\,hr^{-1}}$ or more.

As expected, regions of high $NO_2$ production in both MOZART and RADM2 simulations are seen

over the high $NO_x$-emission regions including Benelux, southern England, western Germany, the
Po Valley, and major cities including Paris and Moscow. The difference in net $NO_2$ production rate
between the two mechanisms is also highest where the absolute $NO_2$ production rates are highest;
in these areas the net $NO_2$ production rate is lower for MOZART than for RADM2 by greater than
0.25 $\mathrm{ppb\,hr^{-1}}$. Furthermore, areas where the two mechanisms show the greatest differences in net
$NO_2$ production rate tend to be the areas where the net $O_3$ production rate is most different between
the two mechanisms, including the large maximum over the Netherlands and northwest Germany.

To further investigate the differences between ozone chemistry in MOZART vs. RADM2, we

performed two additional sensitivity studies with each mechanism: one in which all anthropogenic
$NO_x$ emissions were increased by 30%, and one in which all anthropogenic VOC emissions are
increased by 30%. We then examined the change in $O_3$ concentrations due to these emission pertur-
bations to diagnose whether the chemical mechanisms are operating in a $NO_x$-sensitive or a VOC-
sensitive regime. Results are shown in Figure 16. For the sensitivities where $NO_x$ emissions were
increased by 30%, MOZART and RADM2 simulations show very similar behavior. Most of the do-
main is $NO_x$ sensitive, with increased $NO_x$ emissions resulting in increased modeled $O_3$. Notably,
the U.K., Benelux, northern France and Paris, and northwest Germany show $NO_x$ titration behav-
ior, in which increased $NO_x$ emissions lead to decreased $O_3$ concentrations. $NO_x$ titration behavior
is also seen around the area of the Mediterranean between Monaco, Genoa and Corsica. Magni-
tudes of the observed change in $O_3$ are quite similar for both mechanisms, although RADM2 shows
slightly stronger $NO_x$ titration in the area centered around Benelux, and stronger $NO_x$ sensitivity
over Scandinavia and northwest Russia.

In contrast to the similar behavior seen for $NO_x$ sensitivity, the VOC sensitivity exhibited by

the mechanisms is quite different (Figure 16, lower panel). For both MOZART and RADM2, the
effect of increased anthropogenic VOC emissions on $O_3$ is smaller than the effect of increased $NO_x$
emissions. The MOZART simulation shows very little impact of increased VOC emissions on $O_3$,
with differences in average $O_3$ concentration generally confined to $\pm$ 2% of the base simulation. In
contrast, increasing VOC emissions in the RADM2 simulations leads to increased $O_3$ concentrations



throughout nearly the entire domain. However, the increase in $O_3$ concentration is modest, generally
limited to increases of 2-4% over the base simulation.
Taken as a whole, Figure 16 shows that MOZART behaves in a classically $NO_x$-sensitive manner
for most of domain, with $O_3$ responding to changes in $NO_x$ but showing little response to changes in
anthropogenic VOC. $NO_x$ titration behavior is also observed, particularly around the area of U.K.,
Benelux, and northern France and Germany. RADM2, on the other hand, exhibits more of a mixed
$NO_x$- VOC-sensitivity for much of the domain. The $NO_x$ sensitivity seen in RADM2 is very similar
to that seen in MOZART, but the response of RADM2 to changes in VOC is much stronger (by about
a factor of two) than observed in MOZART. With the exception of some small areas in the North
and Baltic Sea south of Norway and Sweden, RADM2 predicts $O_3$ increases with VOC increases
throughout the entire domain.
In addition to characterizing mechanism behavior with respect to net photochemical $O_3$ produc-
tion and $NO_x$- and VOC-sensitivity, we evaluate the contribution of other sources that could ex-
plain the large differences in predicted $O_3$ between the MOZART and RADM2 simulations. First,
MOZART uses different rate constants for several inorganic gas phase chemical reactions. To test the
effect of these differences all RADM2 inorganic reaction rates were changed so that they matched
those used in MOZART simulations in the cases where the reactions are the same in both mecha-
nisms (Supplementary Material). The differences in inorganic rate constants between the two mech-
anisms explain a significant difference in predicted $O_3$ concentrations: when RADM2 is run with
inorganic rate constants from MOZART, the resulting domain-mean $O_3$ is higher by more than
$8\,\mu g\,m^{-3}$ for the month of July, approximately 40% of the difference in predicted $O_3$.
Besides the gas-phase chemistry itself, there are some differences in the implementation of MOZART-
4 vs. RADM2 in WRF-Chem that could also contribute to the observed differences in modeled $O_3$:
in particular, in the treatment of dry deposition and photolysis (described in the Supplementary Ma-
terial). To test the effect of differences in treatment of dry deposition, we conducted an additional
sensitivity in which we modified the RADM2 simulation to treat dry deposition in the same way as
it is treated in MOZART. However, this led to only a small difference in average ozone (an increase
of $1\,\mu g\,m^{-3}$), indicating that modeled surface $O_3$ concentrations are relatively insensitive to these
differences in the treatment of dry deposition, at least in the summer. In a sensitivity test where we
modified the model code so that the MOZART simulation ran with the same photolysis scheme as
used in our RADM2 simulation (i.e., with the Madronich TUV scheme and without reading in cli-
matological $O_3$ and $O_2$ columns), we found that average $O_3$ for July decreases by $3\,\mu g\,m^{-3}$. This
indicates that modeled $O_3$ is also somewhat sensitive to differences in the treatment of photolysis
in MOZART and RADM2. However, taken together, our sensitivity simulations suggest that the dif-
ferences in the inorganic reaction rate coefficients are more impoprtant than the differing treatments
of dry deposition and photolysis in explaining the differences in predicted $O_3$ between the RADM2
and MOZART simulations.



## 5 Summary and Conclusions


In this paper, we present a detailed description of a WRF-Chem setup over the European domain
and provide an evaluation of the simulated meteorological and chemical fields with an emphasis
on model's ability to reproduce the spatial and temporal distribution of ground-level $O_3$ and $NO_x$.
Within WRF-Chem we compare the performance of two different chemical mechanisms: MOZART-
4, for which we present the first model evaluation for a European domain, and RADM2. Overall, we
found that our WRF-Chem setup reproduced the spatial and seasonal variations in the meteorological
parameters over Europe, with biases and correlations consistent with previous studies. Simulations
using the MOZART-4 as well as RADM2 chemical mechanisms were found to reproduce the spatial
and temporal distributions in ground-level $O_3$ over Europe, based on observations from the EMEP
and Airbase networks. However, we find significant differences in $O_3$ concentrations predicted by the
two chemical mechanisms, with RADM2 predicting as much as $20\,\mu g\,m^{-3}$ less $O_3$ than MOZART
during the spring and summer seasons. In general, MOZART-4 chemistry overpredicts $O_3$ concen-
trations for most of Europe in the summer and fall, whereas RADM2 leads to an underestimation of
$O_3$ over the European domain in all seasons. Taken as a whole, use of MOZART-4 chemistry per-
forms better, leading to lower absolute model biases in $O_3$. This is the case when considering hourly
$O_3$ concentrations as well as metrics relevant for human health, such as MDA8 and SOMO35. De-
spite the large differences in predicted $O_3$, the two mechanisms show relatively similar behavior for
$NO_x$, with both MOZART and RADM2 simulations resulting in a slight underestimation of $NO_x$
compared to surface observations.
The net midday photochemical production rate of $O_3$ in summer is found to be higher for MOZART
than for RADM2 for most of the domain, with the largest differences between the mechanisms seen
over Germany, where the net $O_3$ photochemical production for MOZART is higher than for RADM2
by greater than $1.8\,ppb\,hr^{-1}$ ($3.6\,\mu g\,m^{-3}\,hr^{-1}$). However, we have shown that RADM2 is approx-
imately twice as sensitive to increases in anthropogenic VOC emissions as MOZART, suggesting
that, under local VOC-limited conditions not seen at the regional scale of our simulations, RADM2
is likely to produce $O_3$ at a greater rate than MOZART. Despite the differences in sensitivity to
changes in VOC emissions exhibited by the two mechanisms, sensitivity to changes in $NO_x$ emis-
sions in MOZART and RADM2 are found to be similar.
Our results indicate that modeled surface $O_3$ over Europe is sensitive the choice of gas phase
chemical mechanism, with observed differences in $O_3$ between mechanisms that are larger than
those seen in many past studies. Although the most fundamental differences between MOZART-4
and RADM2 (and other chemical mechanisms used in regional modeling) is the representation of
VOC oxidation chemistry, we find that approximately 40% of the difference seen in predicted $O_3$
seen in this study can be explained by differences in inorganic reaction rate constants employed by
MOZART-4 and RADM2. Further investigation of chemical mechanism behavior within 3-D models
would be helpful to constrain uncertainties in regional air quality modeling.





## 6 Code availability

The WRF-Chem model is an open-source, publicly available software. The code is being continually improved, with new releases approximately twice per year. WRF-Chem code can be downloaded at (http://www2.mmm.ucar.edu/wrf/users/download/get_source.html). The corresponding author will provide the bug fixes to version 3.5.1 used in this study, described in Section 2.3, upon request.

## Appendix A: Abbreviations and Acronyms

DJF: December-January-February (winter)

EDGAR: Emission Database for Global Atmospheric Research

EEA: European Environmental Agency

EOS: Earth Observing System

GEOS5: Goddard Earth Observing System Model, Version 5

GOCART: Goddard Chemistry Aerosol Radiation and Transport

HTAP: Hemispheric Transport of Air Pollution

JJA: June-July-August (summer)

MADE: Modal Aerosol Dynamics Model for Europe

MAM: March-April-May (spring)

MERRA: Modern Era-Retrospective Analysis for Research and Applications

NCEP: National Centers for Environmental Prediction

NCAR: National Center for Atmospheric Research

SON: September-October-November (fall)

SORGAM: Secondary Organic Aerosol Model

WRF-Chem: Weather Research and Forecasting with Chemistry

## Appendix B: Definitions of statistical quantities

The statistical quantities used for model evaluation are defined below. Let $Obs_i^j$ and $Mod_i^j$ be the observed and modeled quantities at time $i$ and station $j$, respectively. $N_{obs}^j$ represents the number of temporal data points evaluated at station $j$, and $N_{obs}$ represents the total number of data points (each representing a time $i$ and a station $j$) evaluated in the domain.

The Mean Bias (MB) at a specific station (e.g., Figure 5) is calculated as

$$MB^j = \frac{1}{N_{obs}^j} \sum_{i=1}^{N_{obs}^j} Mod_i^j - Obs_i^j$$

and the domain-wide Mean Bias (e.g., Table 5) as

$$MB = \frac{1}{N_{obs}} \sum_{i,j=1}^{N_{obs}} Mod_i^j - Obs_i^j$$





Domain-wide values for Normalized Mean Bias (NMB) and Mean Fractional Bias (MFB) are
calculated analogously.
$$NMB = \frac{\sum_{i=1}^{N_{obs}} Mod_i^j - Obs_i^j}{\sum_{i=1}^{N_{obs}} Obs_i^j}$$
$$MFB = \frac{1}{N_{obs}} \sum_{i,j=1}^{N_{obs}} \frac{Mod_i^j - Obs_i^j}{\frac{Mod_i^j + Obs_i^j}{2}}$$
Temporal correlation between model results and observation is evaluated using the Pearson corre-
lation coefficient ($r$). The value of $r$ is calculated at each station using
$$r^j = \frac{\sum_{i=1}^{N_{obs}^j} \left( Mod_i^j - \overline{Mod^j} \right) \left( Obs_i^j - \overline{Obs^j} \right)}{\sigma_{mod} \times \sigma_{obs}}$$
Here, the numerator represents the covariance between the model and observations, $\overline{Mod^j}$ and
$\overline{Obs^j}$ represent the mean of the model and observations, respectively, and $\sigma$ is the standard deviation.
The domain-wide correlation coefficients (e.g., Table 5) is then calculated as
$$r = \frac{1}{N_j} \sum_{j}^{N_j} r^j$$
where $N_j$ is the total number of stations.
*Acknowledgements.* The authors would like to thank Renate Forkel for valuable discussions regarding the
setup of our WRF-Chem simulation. The authors also thank Jane Coates for sharing her technique for VOC
speciation and valuable discussions regarding chemical mechanisms. We thank TNO for access to the TNO-
MACC II emissions inventory, and Hugo Denier van der Gon for helpful discussions regarding emissions.
The HTAP v2.2 anthropogenic emissions were obtained from http://edgar.jrc.ec.europa.eu/htap_v2/index.php.
The authors thank Christophe Knote and Anna Katinka Petersen for sharing bug fixes for the WRF-Chem
MOZART code. WRF-Chem tools for preprocessing boundary conditions as well as biogenic, fire, and anthro-
pogenic emissions were provided by NCAR (http://www.acom.ucar.edu/wrf-chem/download.shtml). Initial and
boundary conditions for meteorological fields were obtained from ECMWF, http://www.ecmwf.int/en/research/
climate-reanalysis/era-interim. Initial and boundary conditions for chemical fields were from MOZART-4/GEOS5,
provided by NCAR at http://www.acd.ucar.edu/wrf-chem/mozart.shtml. Corine land cover data was obtained
from http://www.eea.europa.eu/data-and-maps/data/corine-land-cover-2006-raster-2. We acknowledge the UK
Met Office for providing the Global Weather Observation dataset via the British Atmospheric Data Centre. We
acknowledge EMEP and the Norwegian Institute for Air Research (NILU) for providing the EMEP chemical ob-
servation data via the EBAS database (ebas.nilu.no). AirBase is the public air quality database of the EEA; data
were obtained at http://www.eea.europa.eu/data-and-maps/data/airbase-the-european-air-quality-database-7. The
WRF-Chem simulations have been performed on the supercomputer HYDRA (http://www.rzg.mpg.de/).



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





**Table 1.** WRF-Chem options used in model simulations.

| Atmospheric Process | Option used |
| --- | --- |
| Cloud microphysics | Lin et al. scheme (Lin et al., 1983) |
| Longwave radiation | RRTMG (Iacono et al., 2008) |
| Shortwave radiation | Goddard shortwave scheme (Chou and Suarez, 1994) |
| Surface Layer | MM5 Similarity based on Monin-Obukhov scheme (Beljaars, 1995) |
| Land-surface Physics | Noah Land Surface Model (Chen and Dudhia, 2001) |
| Urban surface physics | Urban Canopy Model (Kusaka and Kimura, 2004) |
| Planetary boundary layer | Yonsei University scheme (Hong et al., 2006) |
| Cumulus parametrization | Grell 3D scheme (Grell and Dévényi, 2002) |

**Table 2.** Description of WRF-Chem simulations performed for this study.

| | Simulation Name | Model Chemistry | Photolysis Scheme |
| --- | --- | --- | --- |
| (1) | MOZART | MOZART-4 chemistry with gocart aerosols, KPP solver | Madronich F-TUV photolysis |
| (2) | RADM2 | RADM2 chemistry with MADE/SORGAM aerosols, KPP solver | Madronich photolysis (TUV) |

**Table 3.** Observational datasets used for model evaluation.

| Database | Parameter | Temporal Resolution | Data Source |
| --- | --- | --- | --- |
| BADC Global Weather Observation Data | MSLP, T2, WS10, WD10 | 3-hourly | http://badc.nerc.ac.uk/home/ |
| AirBase v7 | $O_3$, $NO_2$, NO, $NO_x$ | hourly | http://www.eea.europa.eu/data-and-maps/data/airbase-the-european-air-quality-database-7 |
| EMEP | $NO_2$, NO, $NO_x$ | hourly | http://ebas.nilu.no/ |



**Table 4.** Domain-wide statistical performance of WRF-Chem against 3-hourly meteorological observations from BADC. Modeled quantities are from the MOZART simulation.

| | Winter (DJF) | | | | | | | Spring (MAM) | | | | | | |
|---|---|---|---|---|---|---|---|---|---|---|---|---|---|---|
| | Mean-Obs | Mean-Mod | MB | NMB | MFB | r | no. sta-tions | Mean-Obs | Mean-Mod | MB | NMB | MFB | r | no. sta-tions |
| MSLP (hPa) | 1015.41 | 1014.79 | -0.96 | 0.00 | 0.00 | 0.99 | 1297 | 1014.67 | 1014.46 | -0.35 | 0.00 | 0.00 | 0.99 | 1295 |
| T2 (° C) | 2.51 | 2.99 | 0.29 | 0.11 | -0.01 | 0.89 | 1581 | 9.73 | 9.91 | -0.11 | -0.01 | 0.07 | 0.94 | 1581 |
| WS10 (m/s) | 4.31 | 5.60 | 1.34 | 0.31 | 0.42 | 0.71 | 1577 | 3.86 | 4.46 | 0.65 | 0.17 | 0.29 | 0.68 | 1589 |
| WD10 (deg) | 175.53 | 203.73 | 27.93 | 0.16 | 0.27 | 0.50 | 1568 | 167.88 | 188.67 | 21.16 | 0.13 | 0.25 | 0.48 | 1580 |
| | Summer (JJA) | | | | | | | Fall (SON) | | | | | | |
| | Mean-Obs | Mean-Mod | MB | NMB | MFB | r | no. sta-tions | Mean-Obs | Mean-Mod | MB | NMB | MFB | r | no. sta-tions |
| MSLP (hPa) | 1012.12 | 1012.11 | 0.04 | 0.00 | 0.00 | 0.98 | 1288 | 1017.61 | 1017.42 | -0.49 | 0.00 | 0.00 | 0.99 | 1297 |
| T2 (° C) | 17.82 | 17.70 | -0.38 | -0.02 | 0.00 | 0.87 | 1573 | 9.20 | 9.65 | 0.24 | 0.03 | -0.08 | 0.95 | 1583 |
| WS10 (m/s) | 3.45 | 3.90 | 0.48 | 0.14 | 0.27 | 0.63 | 1574 | 3.64 | 4.61 | 1.04 | 0.28 | 0.40 | 0.68 | 1585 |
| WD10 (deg) | 173.88 | 196.92 | 23.27 | 0.13 | 0.25 | 0.45 | 1561 | 172.30 | 196.49 | 24.02 | 0.14 | 0.27 | 0.48 | 1574 |



**Table 5.** Statistics for MOZART simulation against hourly observations from the AirBase network. Means and MB are expressed in $\mu g\,m^{-3}$; NMB, MFB, and r are unitless. r is the hourly temporal correlation coefficient for all quantities except MDA8, for which it represents the daily temporal correlation coefficient.

| | Winter (DJF) | | | | | | | Spring (MAM) | | | | | | |
|---|---|---|---|---|---|---|---|---|---|---|---|---|---|---|
| | Mean-Obs | Mean-Mod | MB | NMB | MFB | r | no. stations | Mean-Obs | Mean-Mod | MB | NMB | MFB | r | no. stations |
| $O_3$ | 53.82 | 48.34 | -5.44 | -0.10 | -0.10 | 0.60 | 366 | 75.26 | 70.93 | -4.25 | -0.06 | -0.07 | 0.56 | 371 |
| MDA8 | 67.50 | 64.20 | -3.30 | -0.05 | -0.04 | 0.76 | 365 | 96.33 | 97.00 | 0.67 | 0.01 | 0.00 | 0.69 | 370 |
| $NO_x$ | 20.22 | 16.99 | -3.20 | -0.16 | 0.00 | 0.37 | 204 | 14.30 | 13.32 | -0.99 | -0.07 | -0.15 | 0.25 | 210 |
| $NO_2$ | 14.40 | 14.83 | 0.48 | 0.03 | 0.07 | 0.42 | 250 | 11.34 | 12.03 | 0.70 | 0.06 | -0.10 | 0.30 | 252 |
| NO | 4.27 | 1.18 | -3.10 | -0.73 | -1.24 | 0.29 | 148 | 2.65 | 0.79 | -1.87 | -0.70 | -1.26 | 0.27 | 148 |
| | Summer (JJA) | | | | | | | Fall (SON) | | | | | | |
| | Mean-Obs | Mean-Mod | MB | NMB | MFB | r | no. stations | Mean-Obs | Mean-Mod | MB | NMB | MFB | r | no. stations |
| $O_3$ | 70.84 | 80.72 | 9.92 | 0.14 | 0.14 | 0.55 | 370 | 47.24 | 53.10 | 6.14 | 0.13 | 0.13 | 0.57 | 367 |
| MDA8 | 94.51 | 110.37 | 15.86 | 0.17 | 0.16 | 0.61 | 369 | 63.81 | 74.82 | 11.01 | 0.17 | 0.15 | 0.65 | 367 |
| $NO_x$ | 10.63 | 10.57 | -0.10 | -0.01 | -0.21 | 0.16 | 206 | 19.14 | 16.62 | -2.53 | -0.13 | -0.07 | 0.32 | 208 |
| $NO_2$ | 8.30 | 9.66 | 1.37 | 0.17 | -0.12 | 0.22 | 248 | 13.60 | 15.23 | 1.64 | 0.12 | 0.05 | 0.38 | 253 |
| NO | 2.01 | 0.48 | -1.53 | -0.76 | -1.36 | 0.19 | 148 | 4.24 | 1.07 | -3.17 | -0.75 | -1.32 | 0.28 | 146 |



**Table 6.** Statistics for MOZART simulation against hourly observations from the EMEP network. Means and MB are expressed in $\mu g\,m^{-3}$; NMB, MFB, and r are unitless. r is the hourly temporal correlation coefficient for all quantities except MDA8, for which it represents the daily temporal correlation coefficient.

| | Winter (DJF) | | | | | | | Spring (MAM) | | | | | | |
|---|---|---|---|---|---|---|---|---|---|---|---|---|---|---|
| | Mean-Obs | Mean-Mod | MB | NMB | MFB | r | no. stations | Mean-Obs | Mean-Mod | MB | NMB | MFB | r | no. stations |
| O$_3$ | 54.54 | 43.82 | -10.46 | -0.19 | -0.22 | 0.53 | 118 | 78.99 | 68.62 | -10.53 | -0.13 | -0.16 | 0.55 | 120 |
| MDA8 | 64.66 | 55.09 | -9.57 | -0.15 | -0.16 | 0.56 | 117 | 95.64 | 90.15 | -5.49 | -0.06 | -0.07 | 0.65 | 119 |
| NO$_x$ | 11.36 | 12.39 | 1.10 | 0.10 | 0.18 | 0.42 | 8 | 10.21 | 10.44 | 0.41 | 0.04 | -0.04 | 0.33 | 9 |
| NO$_2$ | 10.19 | 13.24 | 3.09 | 0.30 | 0.25 | 0.53 | 34 | 8.07 | 10.72 | 2.55 | 0.32 | -0.01 | 0.37 | 38 |
| NO | 2.10 | 1.22 | -0.87 | -0.41 | -0.65 | 0.36 | 25 | 1.34 | 0.78 | -0.56 | -0.42 | -0.50 | 0.35 | 27 |
| | Summer (JJA) | | | | | | | Fall (SON) | | | | | | |
| | Mean-Obs | Mean-Mod | MB | NMB | MFB | r | no. stations | Mean-Obs | Mean-Mod | MB | NMB | MFB | r | no. stations |
| O$_3$ | 72.08 | 76.39 | 4.04 | 0.06 | 0.06 | 0.54 | 120 | 53.24 | 52.05 | -1.08 | -0.02 | -0.02 | 0.54 | 122 |
| MDA8 | 91.24 | 101.48 | 10.24 | 0.11 | 0.11 | 0.59 | 119 | 66.99 | 70.37 | 3.39 | 0.05 | 0.04 | 0.57 | 121 |
| NO$_x$ | 7.62 | 8.44 | 0.94 | 0.12 | -0.12 | 0.30 | 9 | 11.83 | 12.14 | 0.76 | 0.06 | 0.03 | 0.34 | 9 |
| NO$_2$ | 6.07 | 9.10 | 2.96 | 0.49 | 0.06 | 0.30 | 38 | 8.88 | 13.81 | 5.08 | 0.57 | 0.23 | 0.40 | 38 |
| NO | 1.23 | 0.60 | -0.64 | -0.52 | -0.52 | 0.28 | 29 | 1.42 | 1.23 | -0.14 | -0.10 | -0.36 | 0.34 | 28 |

**Table 7.** Statistics for yearly SOMO35 in $mg\,m^{-3} \cdot days$.

| Simulation | Observation network | Obs | Model | MB | NMB | MFB | no. stations |
|---|---|---|---|---|---|---|---|
| MOZART | AirBase | 6.23 | 8.22 | 1.98 | 0.32 | 0.30 | 375 |
| MOZART | EMEP | 5.73 | 6.27 | 0.51 | 0.09 | 0.11 | 122 |
| RADM2 | AirBase | 6.23 | 2.55 | -3.68 | -0.59 | -0.87 | 375 |
| RADM2 | EMEP | 5.73 | 1.84 | -3.91 | -0.68 | -1.13 | 122 |



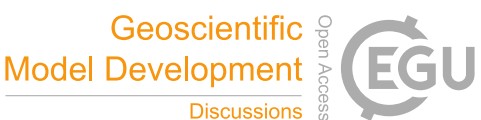

**Table 8.** Statistics for RADM2 simulation against hourly observations from the AirBase network. Means and MB are expressed in $\mu g\,m^{-3}$; NMB, MFB, and r are unitless. r is the hourly temporal correlation coefficient for all quantities except MDA8, for which it represents the daily temporal correlation coefficient.

| | Winter (DJF) | | | | | | | Spring (MAM) | | | | | | |
|---|---|---|---|---|---|---|---|---|---|---|---|---|---|---|
| | Mean-Obs | Mean-Mod | MB | NMB | MFB | r | no. stations | Mean-Obs | Mean-Mod | MB | NMB | MFB | r | no. stations |
| $O_3$ | 53.82 | 41.57 | -12.18 | -0.23 | -0.25 | 0.60 | 366 | 75.26 | 53.36 | -21.81 | -0.29 | -0.33 | 0.53 | 371 |
| MDA8 | 67.50 | 56.04 | -11.46 | -0.17 | -0.17 | 0.75 | 365 | 96.33 | 74.73 | -21.60 | -0.22 | -0.25 | 0.67 | 370 |
| $NO_x$ | 20.22 | 13.75 | -6.45 | -0.32 | -0.23 | 0.36 | 204 | 14.30 | 11.44 | -2.87 | -0.20 | -0.32 | 0.21 | 210 |
| $NO_2$ | 14.40 | 11.90 | -2.47 | -0.17 | -0.15 | 0.41 | 250 | 11.34 | 10.31 | -1.01 | -0.09 | -0.27 | 0.27 | 252 |
| NO | 4.27 | 0.97 | -3.31 | -0.77 | -1.34 | 0.27 | 148 | 2.65 | 0.67 | -1.99 | -0.75 | -1.34 | 0.26 | 148 |
| | Summer (JJA) | | | | | | | Fall (SON) | | | | | | |
| | Mean-Obs | Mean-Mod | MB | NMB | MFB | r | no. stations | Mean-Obs | Mean-Mod | MB | NMB | MFB | r | no. stations |
| $O_3$ | 70.84 | 57.79 | -13.01 | -0.18 | -0.18 | 0.58 | 370 | 47.24 | 39.00 | -8.03 | -0.17 | -0.18 | 0.59 | 367 |
| MDA8 | 94.51 | 80.59 | -13.92 | -0.15 | -0.15 | 0.71 | 369 | 63.81 | 56.02 | -7.79 | -0.12 | -0.12 | 0.69 | 367 |
| $NO_x$ | 10.63 | 9.79 | -0.87 | -0.08 | -0.29 | 0.14 | 206 | 19.14 | 14.30 | -4.84 | -0.25 | -0.24 | 0.30 | 208 |
| $NO_2$ | 8.30 | 8.95 | 0.67 | 0.08 | -0.19 | 0.21 | 248 | 13.60 | 12.57 | -1.01 | -0.07 | -0.13 | 0.36 | 253 |
| NO | 2.01 | 0.46 | -1.55 | -0.77 | -1.42 | 0.18 | 148 | 4.24 | 1.28 | -2.97 | -0.70 | -1.27 | 0.26 | 146 |





**Table 9.** Statistics for RADM2 simulation against hourly observations from the EMEP network. Means and MB are expressed in $\mu g\, m^{-3}$; NMB, MFB, and r are unitless. r is the hourly temporal correlation coefficient for all quantities except MDA8, for which it represents the daily temporal correlation coefficient.

| | Winter (DJF) | | | | | | | Spring (MAM) | | | | | | |
|---|---|---|---|---|---|---|---|---|---|---|---|---|---|---|
| | Mean-Obs | Mean-Mod | MB | NMB | MFB | r | no. stations | Mean-Obs | Mean-Mod | MB | NMB | MFB | r | no. stations |
| O$_3$ | 54.54 | 38.67 | -15.62 | -0.29 | -0.36 | 0.54 | 118 | 78.99 | 53.24 | -25.83 | -0.33 | -0.40 | 0.49 | 120 |
| MDA8 | 64.66 | 49.40 | -15.26 | -0.24 | -0.27 | 0.56 | 117 | 95.64 | 71.04 | -24.60 | -0.26 | -0.29 | 0.55 | 119 |
| NO$_x$ | 11.36 | 10.31 | -0.99 | -0.09 | -0.02 | 0.38 | 8 | 10.21 | 8.76 | -1.31 | -0.13 | -0.24 | 0.30 | 9 |
| NO$_2$ | 10.19 | 10.72 | 0.56 | 0.06 | 0.03 | 0.51 | 34 | 8.07 | 9.11 | 0.95 | 0.12 | -0.19 | 0.34 | 38 |
| NO | 2.10 | 1.16 | -0.93 | -0.44 | -0.67 | 0.37 | 25 | 1.34 | 0.68 | -0.67 | -0.50 | -0.59 | 0.31 | 27 |
| | Summer (JJA) | | | | | | | Fall (SON) | | | | | | |
| | Mean-Obs | Mean-Mod | MB | NMB | MFB | r | no. stations | Mean-Obs | Mean-Mod | MB | NMB | MFB | r | no. stations |
| O$_3$ | 72.08 | 55.65 | -16.65 | -0.23 | -0.24 | 0.58 | 120 | 53.24 | 39.89 | -13.21 | -0.25 | -0.29 | 0.57 | 122 |
| MDA8 | 91.24 | 74.75 | -16.49 | -0.18 | -0.19 | 0.69 | 119 | 66.99 | 54.31 | -12.68 | -0.19 | -0.21 | 0.63 | 121 |
| NO$_x$ | 7.62 | 7.61 | 0.10 | 0.01 | -0.24 | 0.28 | 9 | 11.83 | 10.59 | -0.82 | -0.07 | -0.13 | 0.32 | 9 |
| NO$_2$ | 6.07 | 8.33 | 2.20 | 0.36 | -0.02 | 0.29 | 38 | 8.88 | 11.48 | 2.71 | 0.31 | 0.04 | 0.39 | 38 |
| NO | 1.23 | 0.52 | -0.73 | -0.59 | -0.58 | 0.25 | 29 | 1.42 | 1.43 | 0.07 | 0.05 | -0.31 | 0.31 | 28 |





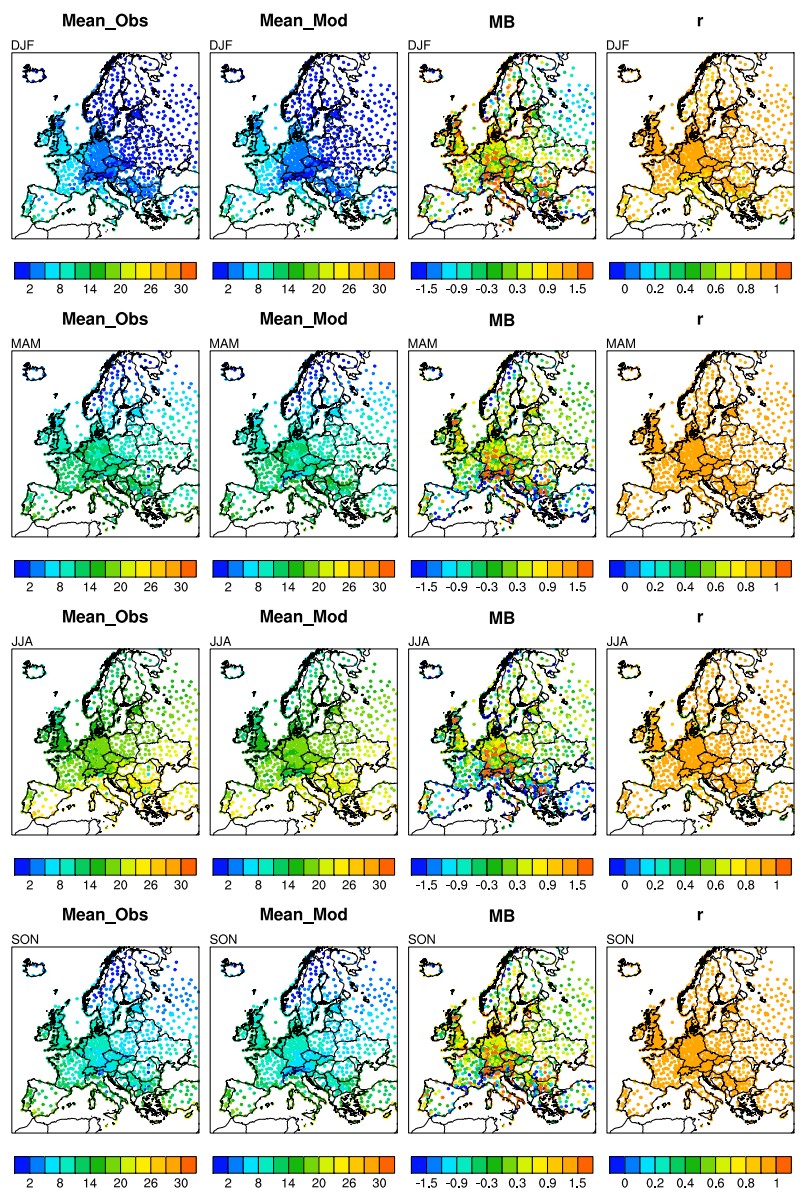

**Figure 1.** Seasonal average values of 2-meter temperature (T2) in degrees Celcius. Model results and statistics are shown for the MOZART simulation at the locations of the observations.

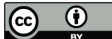



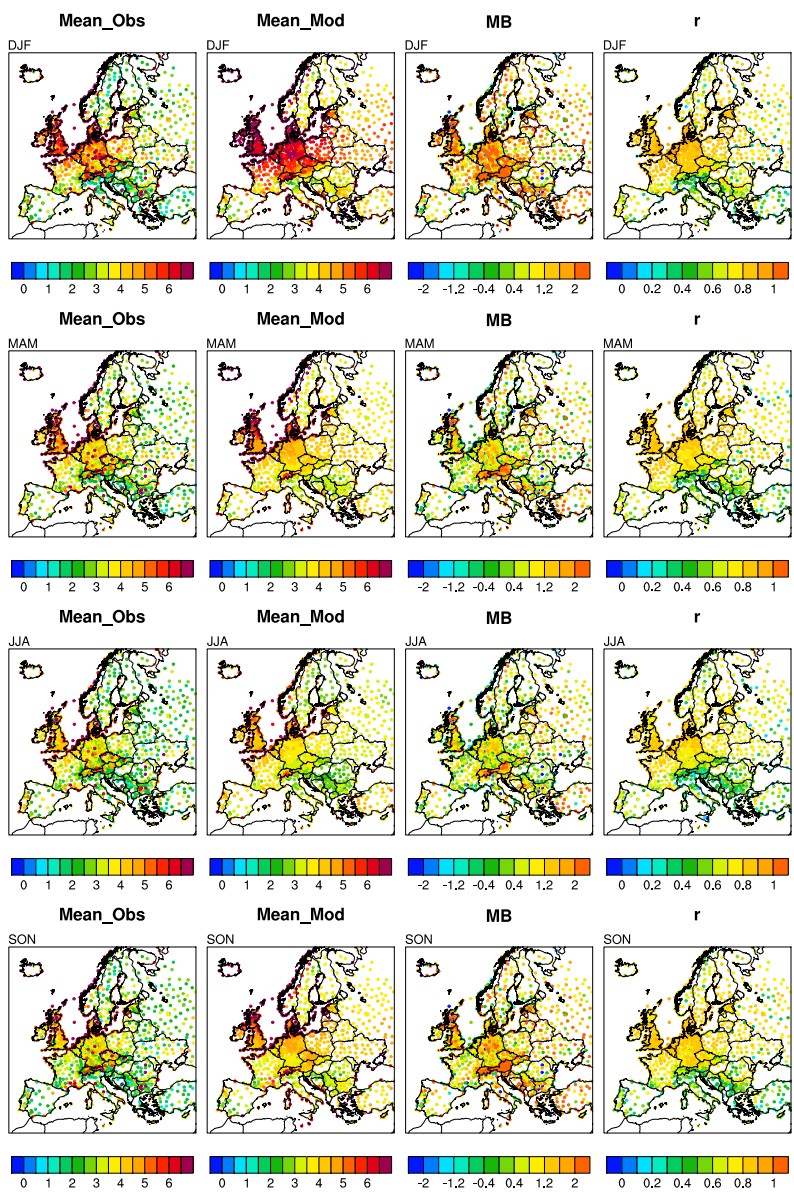

**Figure 2.** Seasonal average values of 10-meter wind speed (WS10) in m/s. Model results and statistics are shown for the MOZART simulation at the locations of the observations.





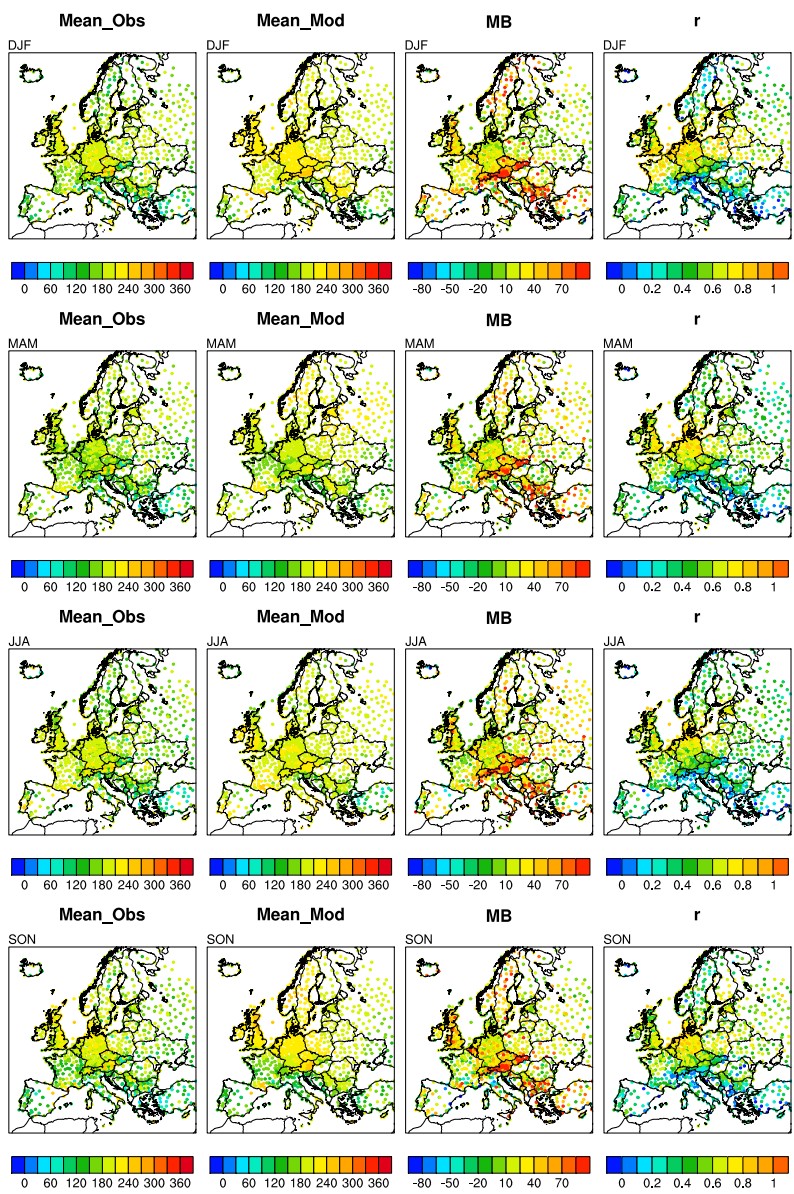

**Figure 3.** Seasonal average values of 10-meter wind speed (WS10) in m/s. Model results and statistics are shown for the MOZART simulation at the locations of the observations.







**Figure 4.** Seasonal average values of surface $O_3$ in $\mu g\,m^{-3}$. Contours are model output from the MOZART simulation. Filled dots represent hourly measurements at AirBase rural background stations, filled squares represent measurements at EMEP stations.





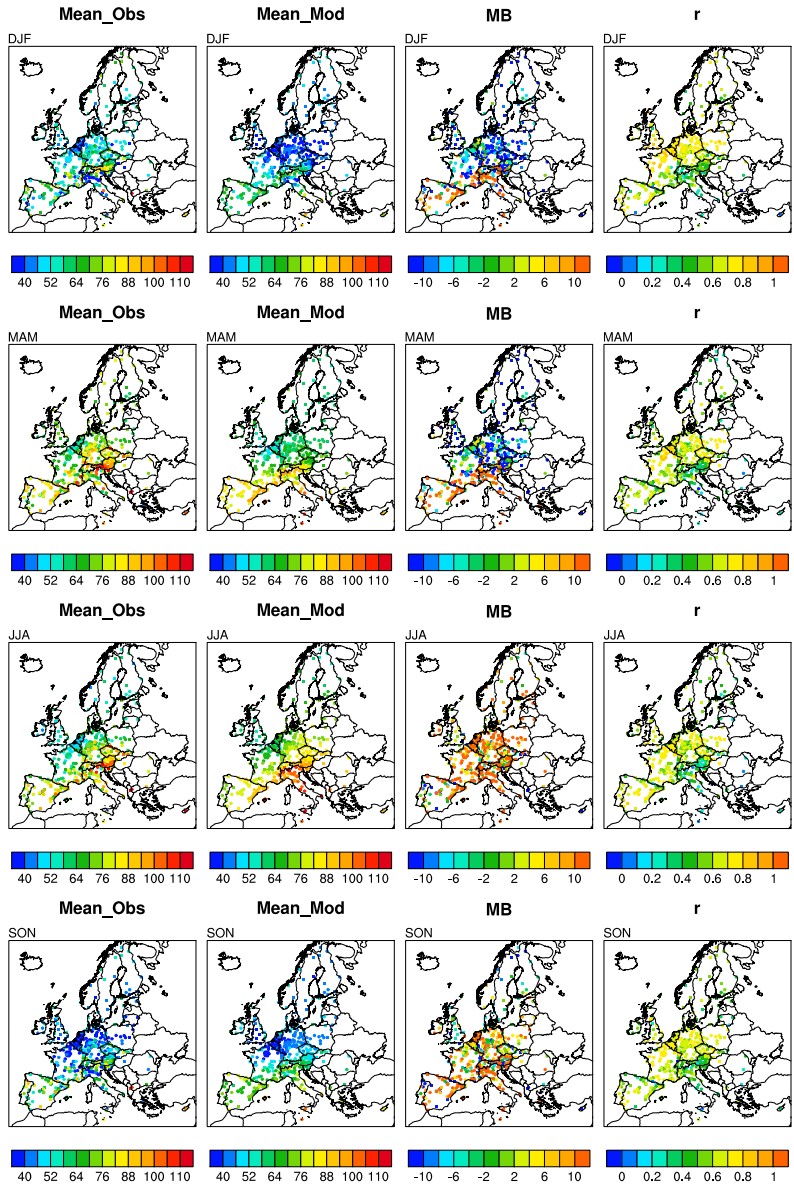

**Figure 5.** Seasonal average values of surface $O_3$ in $\mu g\,m^{-3}$ from hourly measurements at AirBase (circles) and EMEP (squares) stations, and modeled values from MOZART for corresponding locations. The Mean Bias (MB, in $\mu g\,m^{-3}$) and temporal correlation coefficient (r) for hourly values are also shown at the location of station observations.





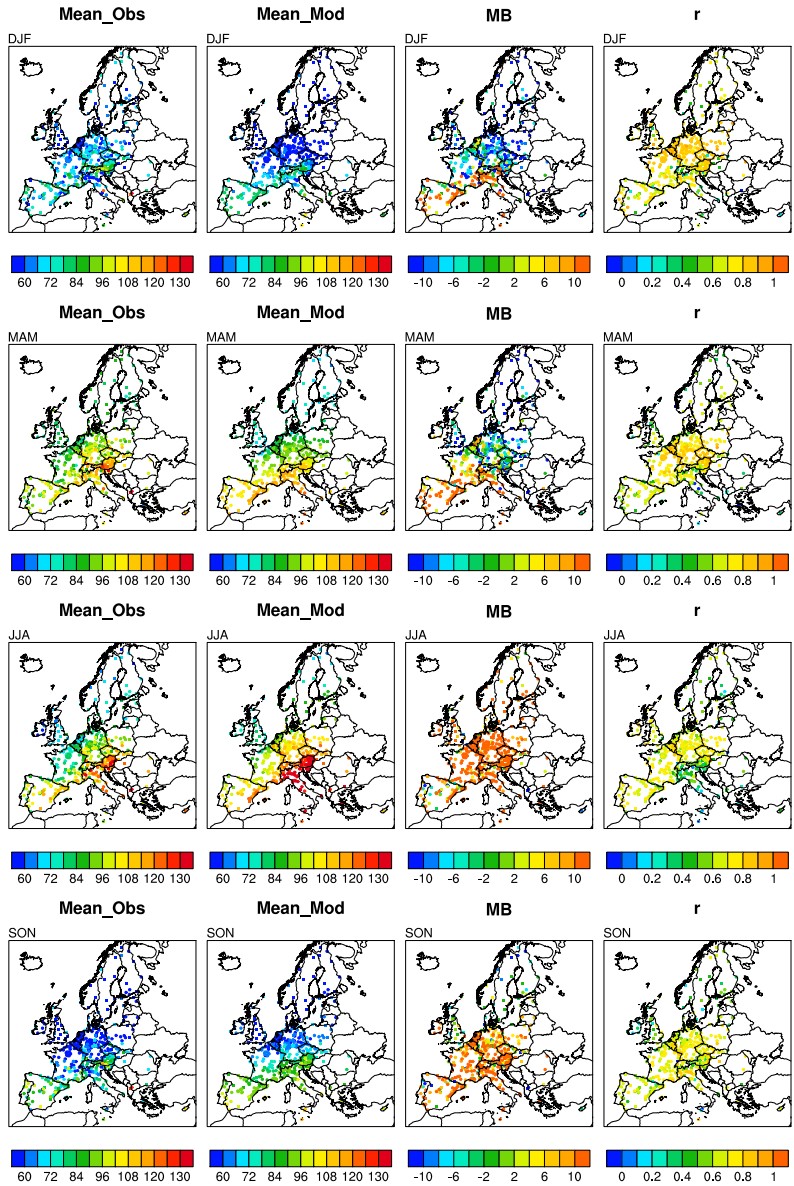

**Figure 6.** Seasonal average values of MDA8 in µg m$^{-3}$ calculated from hourly measurements at AirBase (circles) and EMEP (squares) stations, and modeled values from MOZART for corresponding locations. The Mean Bias (MB, in µg m$^{-3}$) and temporal correlation coefficient (r) for daily values are also shown at the location of station observations.





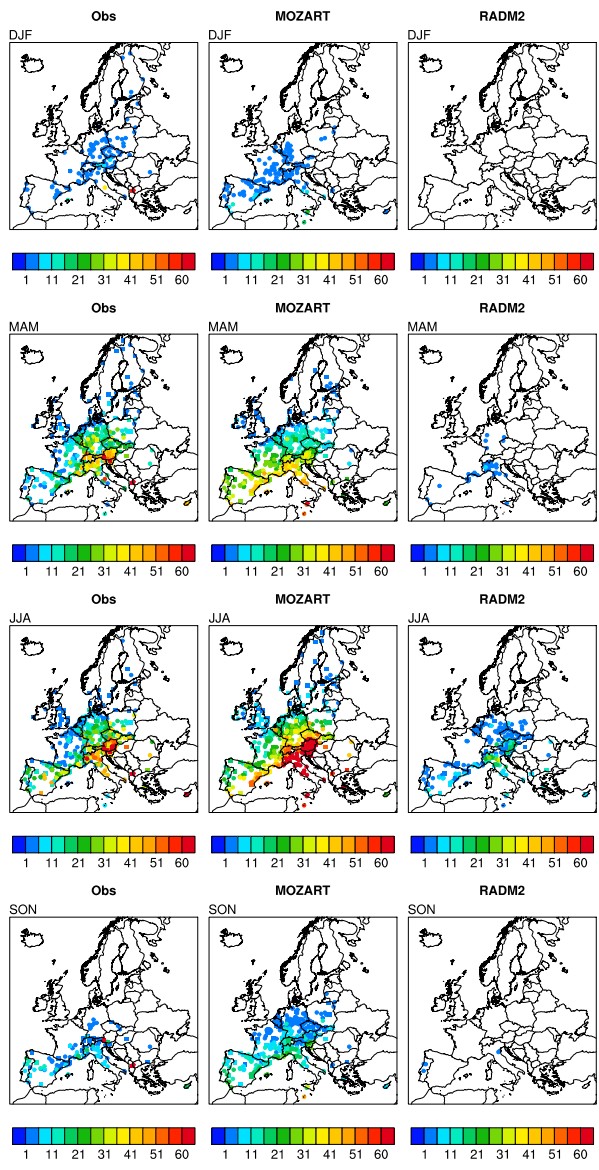

**Figure 7.** Number of days of exceedances of the EU long-term objective value for MDA8 (120 µg m$^{-3}$) at AirBase (circles) and EMEP (squares) station locations. Shown are totals by season for observations and the MOZART and RADM2 simulations. For simplicity of viewing the data, stations with no exceedances are not plotted.



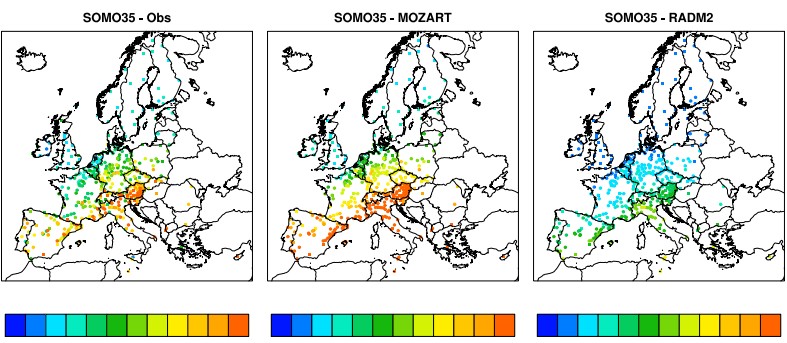

**Figure 8.** Yearly values of SOMO35 in $\mathrm{mg\,m^{-3}} \cdot \mathrm{days}$ calculated from hourly measurements at AirBase (circles) and EMEP (squares) stations, and modeled values for corresponding locations.





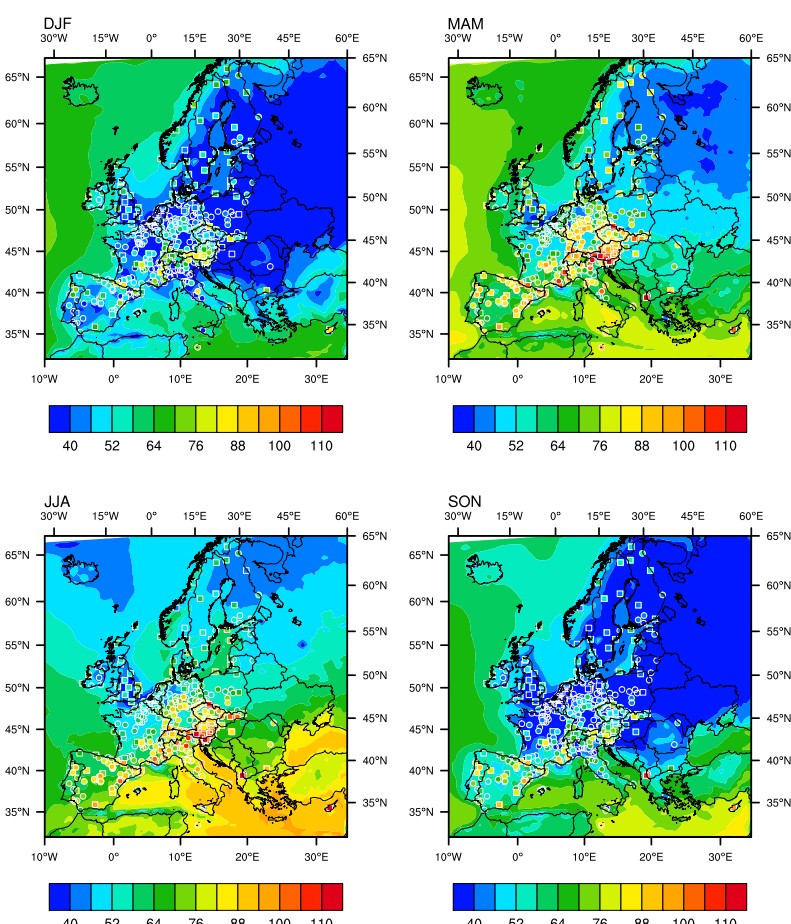

**Figure 9.** Seasonal average values of surface $O_3$ in $\mu g\,m^{-3}$. Contours are model output from the RADM2 simulation. Filled dots represent hourly measurements at AirBase rural background stations, filled squares represent measurements at EMEP stations.





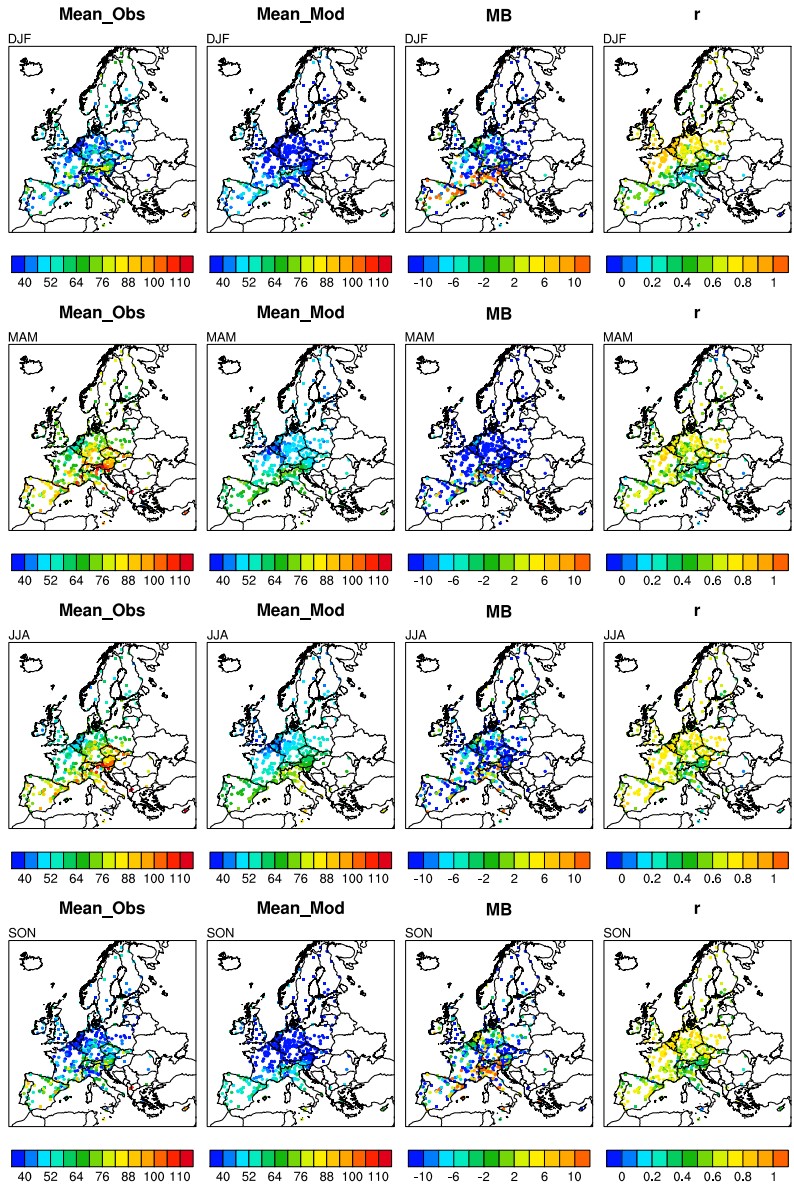

**Figure 10.** Seasonal average values of surface $O_3$ in µg m$^{-3}$ from hourly measurements at AirBase (circles) and EMEP (squares) stations, and modeled values from RADM2 for corresponding locations. The Mean Bias (MB, in µg m$^{-3}$) and temporal correlation coefficient (r) for hourly values are also shown at the location of station observations.





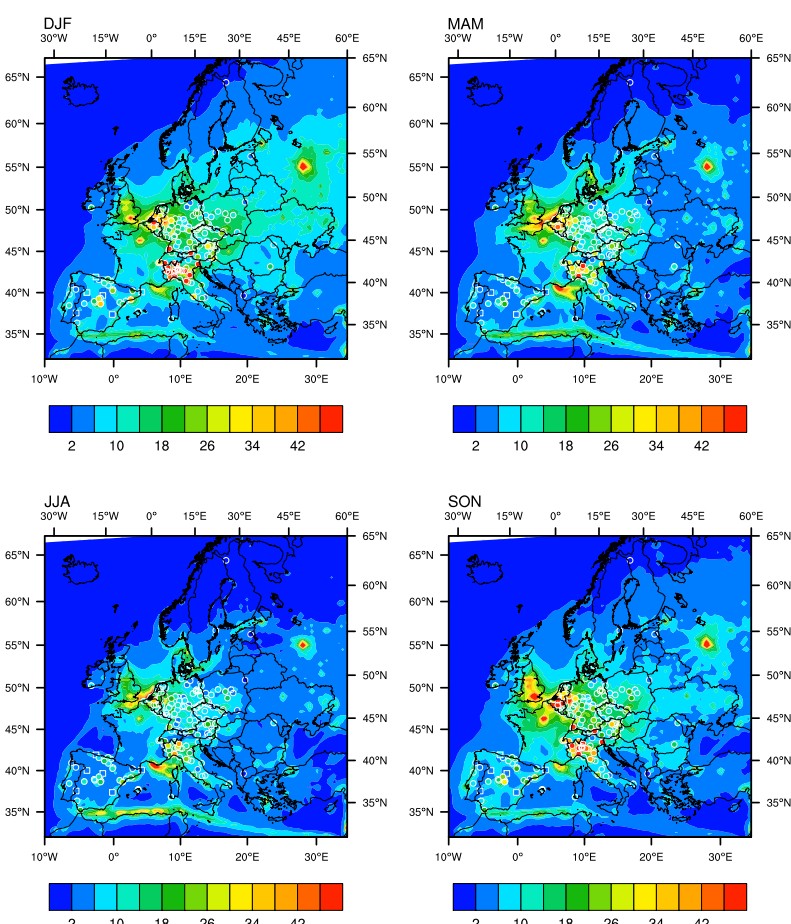

**Figure 11.** Seasonal average values of surface $NO_x$ in $\mu g\,m^{-3}$. Contours are model output from the MOZART simulation. Filled dots represent hourly measurements at AirBase rural background stations, filled squares represent measurements at EMEP stations.





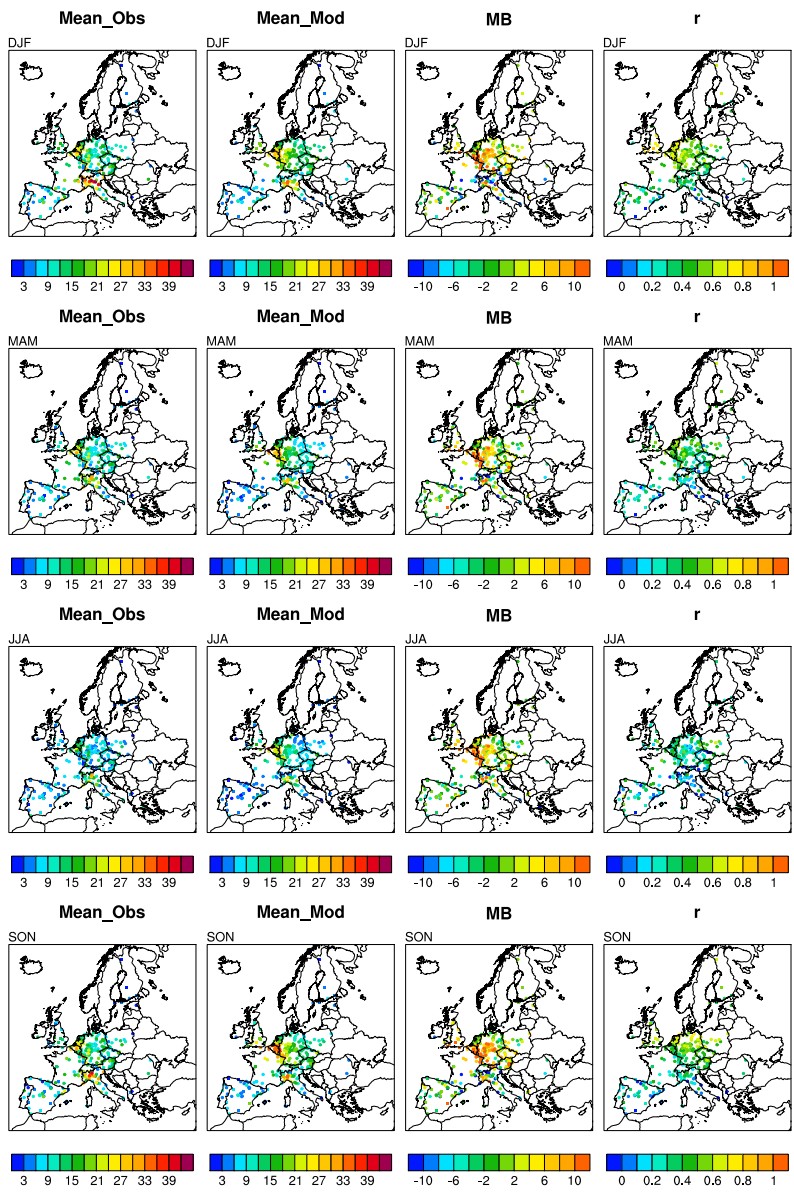

**Figure 12.** Seasonal average values of surface $NO_2$ in $\mu g\,m^{-3}$ from hourly measurements at AirBase (circles) and EMEP (squares) stations, and modeled values from MOZART for corresponding locations. The Mean Bias (MB) and temporal correlation coefficient (r) for hourly values are also shown at the location of station observations.



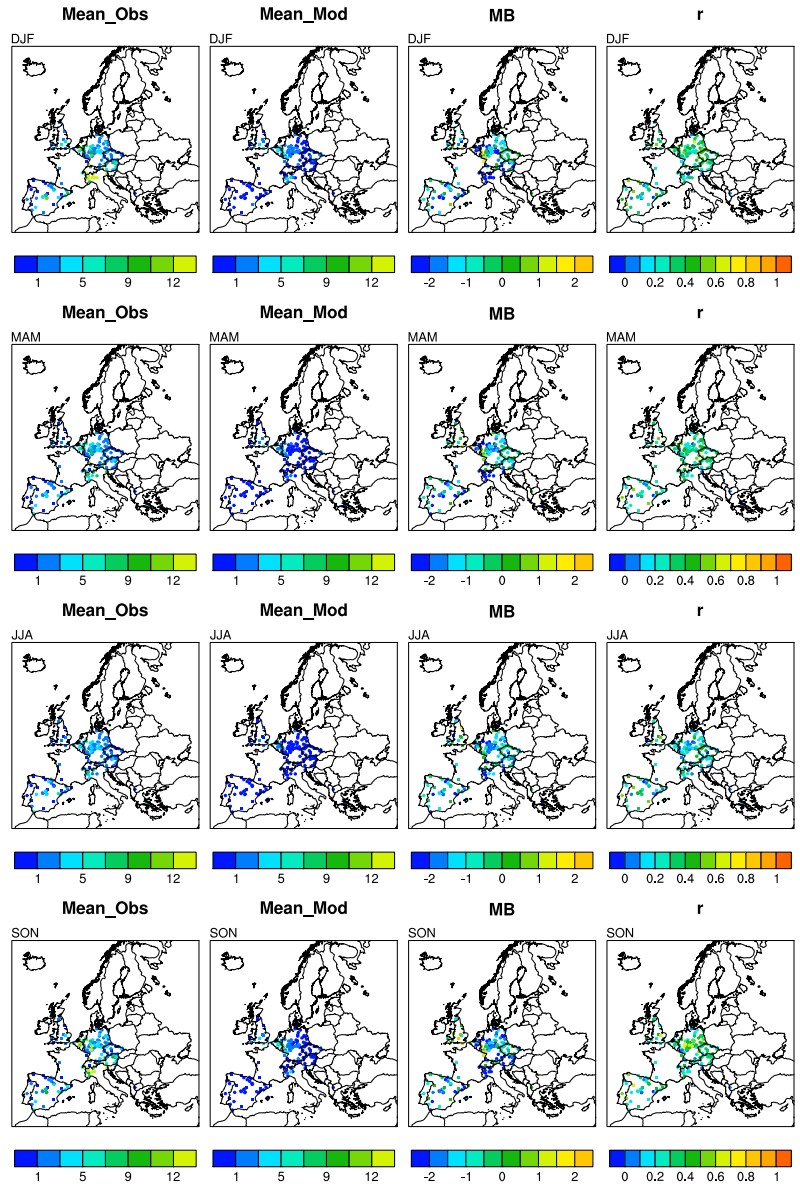

**Figure 13.** Seasonal average values of surface NO in µg m$^{-3}$ from hourly measurements at AirBase (circles) and EMEP (squares) stations, and modeled values from MOZART for corresponding locations. The Mean Bias (MB) and temporal correlation coefficient (r) for hourly values are also shown at the location of station observations.





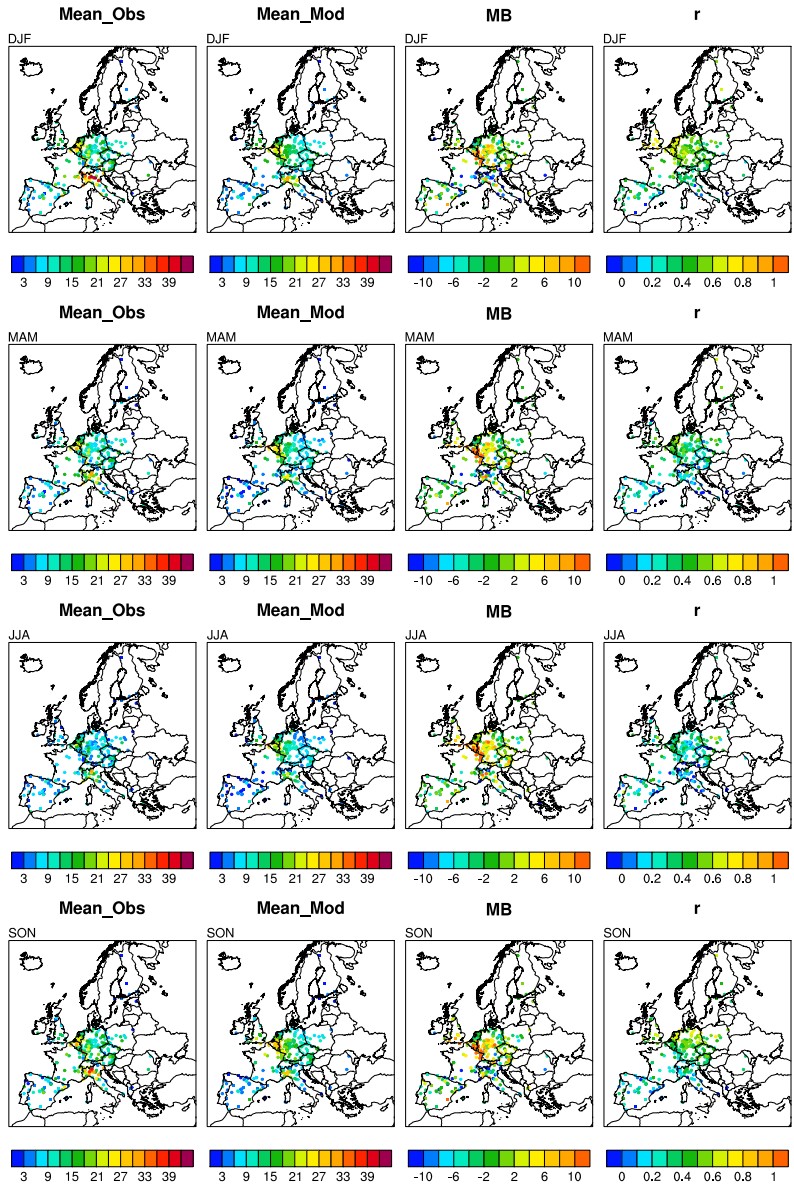

**Figure 14.** Seasonal average values of surface $NO_2$ in $\mu g\,m^{-3}$ from hourly measurements at AirBase (circles) and EMEP (squares) stations, and modeled values from RADM2 for corresponding locations. The Mean Bias (MB) and temporal correlation coefficient (r) for hourly values are also shown at the location of station observations.



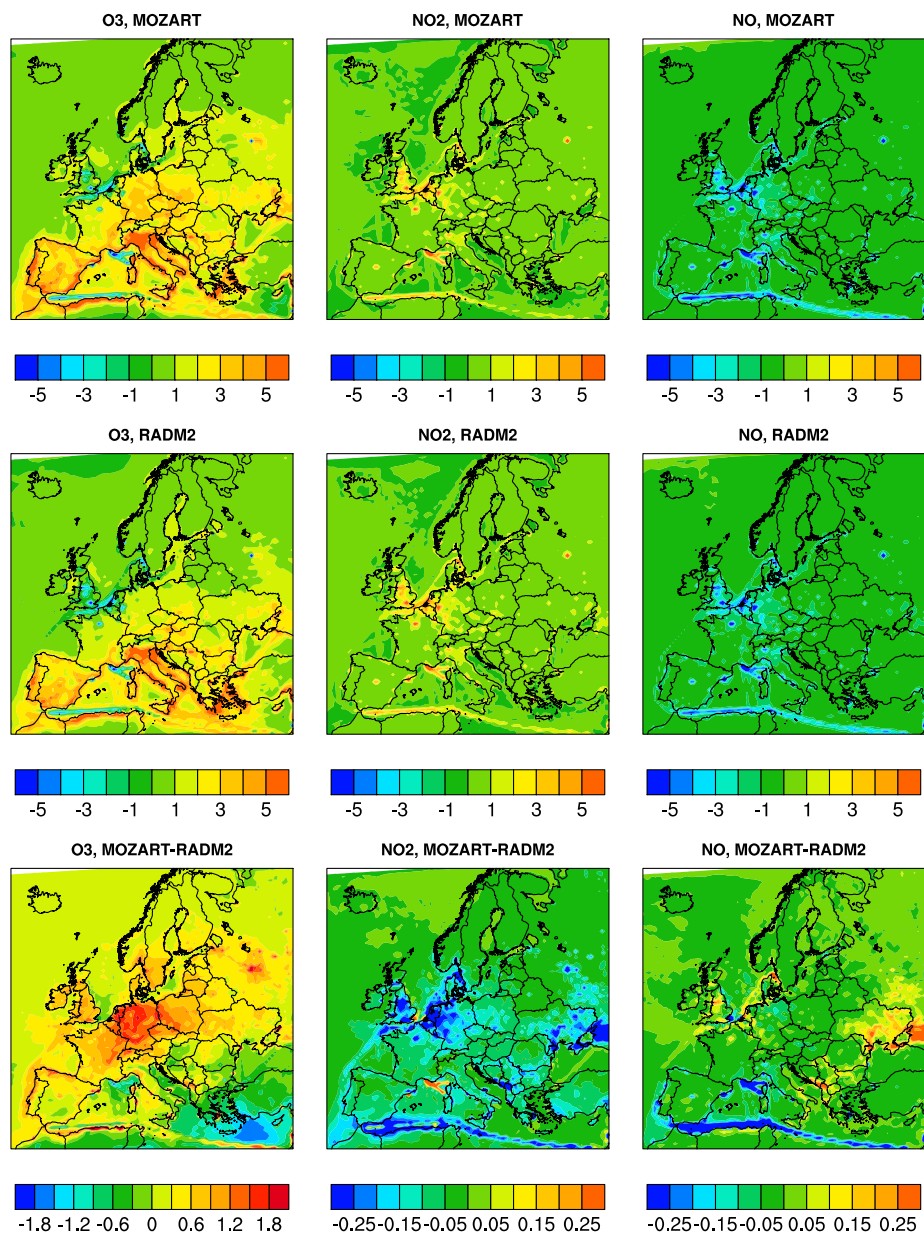

**Figure 15.** Net midday (11:00 - 14:00 CEST) photochemical production rate in $\mathrm{ppb\,hr^{-1}}$ for $O_3$, $NO_2$, and NO shown for MOZART and RADM2 for July 2007. The last row shows the difference in net production rate in $\mathrm{ppb\,hr^{-1}}$ (RADM2 subtracted from MOZART).





**Figure 16.** Sensitivity of average $O_3$ for July 2007 to a 30% increase in emissions of $NO_x$ (upper row) or VOC (lower row), shown for the MOZART and RADM2 chemical mechanisms. Shown here is the fractional change in $O_3$ concentration, i.e., $([O3]_{+30\%emissions}-[O3]_{base})/[O3]_{base}$.