# Peer review of "Ozone air quality simulations with WRF-Chem (v3.5.1) over Europe: Model evaluation and chemical mechanism comparison"

_Geoscientific Model Development, 2016_

## Referee Comment (RC1) · Anonymous Referee #1 · 7 Jul 2016

This manuscript presents the application of the WRF-Chem model over Europe with particular focus on ground level ozone concentrations. Model is evaluated for two different chemical mechanisms (MOZART-4 and RADM2) for the whole year 2007.

The article is well written and the methodology is clearly and very deeply described. The modelling performance evaluation is well treated by the authors as well as an interesting characterization of the chemical mechanism differences is presented too. Moreover, the consideration on poorly analyzed aspects, such as the photochemistry production rates, makes this study an important element of the common effort to evaluate more deeply the quality of the chemistry-transport models and the chemical mechanisms themselves.

For these reasons, I consider that such work should be published in GMD, but only after minor revisions are addressed:

Page 3 line 54-55: I disagree with this statement. Many European WRF-Chem modelling evaluation studies have been published in the last few years.

Page 10 line 293-295: Please, in order to prove that differences between the two meteorological simulations are negligible provide statistical indexes or a comparison figure in the supplementary material.

Page 15 line 478: to be in line with the NOx (NO2 and NO) treatment in MOZART simulation, I suggest to briefly explore NO concentrations in RADM2

Figure 1: I suggest to represent temperature using the International System unit (K) here and everywhere else in the text.
* * *

---

## Referee Comment (RC2) · Anonymous Referee #2 · 8 Jul 2016

The submitted manuscript provides an evaluation of the online regional model WRF-Chem over Europe using two chemical mechanisms (MOZART-4 and RADM2) for a full year simulation with focus on near surface ozone and nitrogen oxides. The manuscript has an added value for the WRF-Chem community. I suggest acceptance of the manuscript for publication after taking into consideration the following comments.

Comments 1) lines 50-52: The authors give here three examples of air quality models but maybe they could also refer here to the review article of Baklanov et al. (2014) for the online coupled regional meteorology chemistry models in Europe.

2) lines 62-64: The importance of time variant chemical boundary conditions for simulated near surface ozone over Europe has been also highlighted in other recent regional

modelling studies (see e.g. Akritidis et al., 2013).

3) line 264: Please provide some more information on the selection of the AirBase stations classified as rural background. Do you include stations with class 1–3 according to the Joly-Peuch classification methodology for surface ozone (Joly and Peuch, 2012). This approach has been also applied in a recent study by Katragkou et al. (2015) for the evaluation of MACC reanalysis near-surface ozone over Europe.

4) line 283: You may add one sentence with information for the use and value of SOMO35 index.

5) Looking the Figures 4 and 9 I am wondering why at the lateral boundaries there are such differences between the two simulations with the different chemical mechanisms (RADM2 and MOZART) even though they are constrained with identical O3 chemical lateral boundary conditions.

6) lines 546-551: Normally with NOx titration we mean the first order removal process of O3 through direct reaction with NO which takes place during nighttime and in the vicinity of large NO emission sources. However the presented results refer to summer daytime and maybe this behaviour is related to the saturated NOx conditions (or VOC sensitive conditions) in these areas (which is a different issue). The split between NOx-saturated or NOx-sensitive regimes is driven by the chemistry of odd hydrogen radicals with HNO3 being the dominant sink in the first case and peroxides the dominant sink in the second case. Maybe the authors could also plot the photochemical regimes in their simulations for the month of July using VOC/NOx or H2O2/NOy ratios (see also the study of Beekmann and Vautard, ACP, 2010).

7) lines 558-559: Mind also that the highest sensitivity for ozone production with regards to VOC is at the regions of high NOx emissions as someone would expect for the regions in the VOC limited regime.

8) lines 565-566: Do you think that the different O3 sensitivity to VOC changes in the

two schemes can account for the O3 differences between RADM2 and MOZART (e.g. the lower ozone values in MOZART)? If yes, in which sense?

9) lines 575-578: This is an interesting result which shows that differences in rate constants can account by 40% for the O3 differences between RADM2 and MOZART runs. You may highlight this result a bit more.

10) lines 591-594: Taking into consideration all three (rate constants, deposition and photolysis schemes) it seems that altogether account about 60% for the O3 differences between RADM2 and MOZART runs. Is this correct? You may highlight this conclusion.

11) Figure 3: I guess here the authors refer to wind direction. Please also provide information on the approach calculating the wind direction difference between obs and model.

12) Figure 16: Maybe it would be better to show the sensitivity result in a percentage scale (from -10 to 10 %).

Minor comments line 209: delete double "and". line 239: It is "for" instead of "fo". line 305: Maybe "related" instead of "associated" . line 406: It is "configuration" instead of " configuruation" . lines 427-429: The sentence needs rephrasing. It is not clear.

―――――――――――――――――

---

## Author Comment (AC1) · 28 Aug 2016

The authors would like to thank Anonymous Referee #1 for their constructive comments. Below are our responses.

Page 3 line 54-55: I disagree with this statement. Many European WRF-Chem modelling evaluation studies have been published in the last few years.

We have added additional citations of papers that apply WRF-Chem over Europe. However, in our view, studies that focus on evaluation over the whole European domain are still limited to date. If there are particular studies that fulfill this criteria that are not being discussed in the manuscript, the authors would appreciate it if the referee would

mention the papers specifically. The sentences in question have been updated as follows to improve clarity: "The use of WRF-Chem over Europe has increased in recent years (e.g., Forkel et al., 2012; Žabkar et al., 2015; Solazzo et al., 2012a, b; Tuccella et al., 2012; Zhang et al., 2013a, b). However, only a limited number of these studies are dedicated to the evaluation of WRF-Chem-simulated meteorology and chemistry over the whole European domain."

Page 10 line 293-295: Please, in order to prove that differences between the two meteorological simulations are negligible provide statistical indexes or a comparison figure in the supplementary material.

A table and figures showing the meteorology from the RADM2 simulation has been added to the supplementary material; see Table S1 and Figures S4-S7. Furthermore, the manuscript has been updated as follows to directly address this question.

"Differences in predicted meteorology between the MOZART and RADM2 simulations are small, with differences in MSLP less than one hundredth of 1%, and differences in T2, WS10, and WD10 generally far below 1%. Since the simulations were run without aerosol-radiative feedbacks, it was expected that the two simulations would show minimal differences in meteorology, and we conclude that differences in O3 and NOx predicted in the MOZART and RADM2 simulations (Section 4.2) are a direct result of differences in the chemistry, rather than chemistry-radiative feedbacks. Statistics for meteorology for the RADM2 simulation can be found in the Supplementary Material, Table S1 and Figures S4-S7."

Page 15 line 478: to be in line with the NOx (NO2 and NO) treatment in MOZART simulation, I suggest to briefly explore NO concentrations in RADM2

A discussion of NO concentrations in RADM2 has been added to the revised manuscript, as follows. "Like for MOZART, NO for RADM2 is underpredicted throughout the domain, with NO concentrations slightly more negatively biased than in MOZART in all seasons except Fall, when NO concentrations are higher for RADM2

than for MOZART and show better agreement with the observations. Temporal correlation for NO2 and NO in RADM2 is also found to show similar behavior to the MOZART simulation."

Figure 1: I suggest to represent temperature using the International System unit (K) here and everywhere else in the text.

The authors prefer to keep temperature in units of Celsius. Although it is not the SI unit, Celsius is widely used in the meteorological community, and is also used in GMD publications (see, e.g., http://www.geosci-model-dev.net/9/1959/2016/gmd-9-1959-2016.pdf). Furthermore, when calculating relative bias statistics (MB, NMB, MFB) for temperature as in Table 4, using Kelvin rather than Celsius makes the denominator extremely large and the bias extremely small, making relative bias statistics less meaningful. However, if the editor agrees that the temperature unit should be changed to Kelvin, we will make these changes to our manuscript.

---

## Author Comment (AC2) · 28 Aug 2016

The authors would like to thank Anonymous Referee #2 for their constructive comments. Below are our responses.

1) lines 50-52: The authors give here three examples of air quality models but maybe they could also refer here to the review article of Baklanov et al. (2014) for the online coupled regional meteorology chemistry models in Europe.

The original manuscript did include a citation to Baklanov et al. 2014 (line 53-54). However, we have added in the revised manuscript a more detailed reference to this manuscript: "The application of online coupled regional meteorology chemistry models

in Europe, among them WRF-Chem, has been recently reviewed by Baklanov et al. [2014]."

2) lines 62-64: The importance of time variant chemical boundary conditions for simulated near surface ozone over Europe has been also highlighted in other recent regional modelling studies (see e.g. Akritidis et al., 2013).

Following the referee's suggestion, the manuscript has been extended, mentioning a the importance of temporally varying chemical boundary conditions.

"The importance of temporally varying chemical boundary conditions in air quality modeling has also been stressed in other studies (including Akritidis et al., 2013; Andersson et al., 2015)."

3) line 264: Please provide some more information on the selection of the AirBase stations classified as rural background. Do you include stations with class 1–3 according to the Joly-Peuch classification methodology for surface ozone (Joly and Peuch, 2012). This approach has been also applied in a recent study by Katragkou et al. (2015) for the evaluation of MACC reanalysis near-surface ozone over Europe.

We used the classification of stations provided with the metadata in AirBase. This is now indicated in the revised manuscript in Section 3.2.2.

"Because of the relatively coarse horizontal resolution in this model study, model output is only compared against AirBase stations that are classified as "rural background." The station classification was taken from the metadata provided by the EEA for AirBase."

4) line 283: You may add one sentence with information for the use and value of SOMO35 index.

A brief discussion of the purpose and use of the SOMO35 metric has been added to the manuscript in Section 3.3.

"SOMO35 is an indicator of cumulative annual ozone exposure used in health impact

assessments. The accumulated health impact is assumed to be proportional to the sum of concentrations above a cutoff of 35 ppb, chosen because the relationship between O3 and adverse effects is very uncertain below this threshhold (WHO, 2013)."

5) Looking the Figures 4 and 9 I am wondering why at the lateral boundaries there are such differences between the two simulations with the different chemical mechanisms (RADM2 and MOZART) even though they are constrained with identical O3 chemical lateral boundary conditions.

The importance of ozone import into the model domain from the lateral boundary conditions depends not only the concentration at the lateral boundary conditions (as the reviewer notes, in the case of MOZART and RADM2 simulations, these concentrations are the same), but also on the dominant wind flows at the edge of the domain. A plot of seasonally averaged wind vectors from ERA-Interim for 2007, which are the fields used to force model meteorology at the edges of our domain, has been added to the Supplementary Material (Figure S2). The dominant flow of air onto the European continent is from the west, and we see that the western (particularly northwestern) edge of the domain is where seasonally-averaged O3 values are most similar between the MOZART and RADM2 simulations. At the northwestern edges of the domain, we see that seasonal average O3 predicted by RADM2 is generally not more than 5% lower than that predicted by MOZART. At the southern and eastern edges of the domain, there is not a strong flow of air into the model domain, which dampens the impact of ozone boundary conditions in this area.

In addition to the addition of Figure S2 to the Supplementary Material, we have made the following addition to the text in Section 4.2.1.

"Absolute O3 concentrations are most similar (i.e., less than 5% different) between the mechanisms near the northwest edges of the domain (see Figures 4 and 9), where the prevailing westerly winds (Supplementary Material, Figure S2) mean that O3 imported from the boundary conditions plays a dominant role."

6) lines 546-551: Normally with NOx titration we mean the first order removal process of O3 through direct reaction with NO which takes place during nighttime and in the vicinity of large NO emission sources. However the presented results refer to summer daytime and maybe this behaviour is related to the saturated NOx conditions (or VOC sensitive conditions) in these areas (which is a different issue). The split between NOxsaturated or NOx-sensitive regimes is driven by the chemistry of odd hydrogen radicals with HNO3 being the dominant sink in the first case and peroxides the dominant sink in the second case. Maybe the authors could also plot the photochemical regimes in their simulations for the month of July using VOC/NOx or H2O2/NOy ratios (see also the study of Beekmann and Vautard, ACP, 2010).

The reviewer is correct; in this discussion the term "NOx titration behavior" has been replaced with "NOx saturated behavior." Regarding plotting chemical indicators for chemical regime, an additional plot showing the indicator CH2O/NOy has been added to the Supplementary Material (Figure S12); a brief discussion of this figure is now included in Section 4.3. A comparison of our results on NOx vs. VOC sensitivity to the findings of Beekman and Vautard (2010) has also been added to the discussion. The revised discussion is copied below.

"Notably, the U.K., Benelux, northern France and Paris, and northwest Germany show NOx-saturated behavior, in which increased NOx emissions lead to decreased O3 concentrations. NOx-saturated regimes are also seen around the area of the Mediterranean between Monaco, Genoa and Corsica. An alternate approach to identify areas of NOx-sensitive vs. NOx-saturated regimes is to use indicator ratios (in the base simulation) following Sillman (1995). We have applied this approach with the indicator ratio CH2O/NOy (Figure S12) and find that areas identified as NOx sensitive using the indicator ratio are the same as those identified using the simulation with +30% NOx emissions. These results are also consistent with the areas of Europe found to be NOx saturated in the model study of Beekmann and Vautard (2010). Magnitudes of the observed change in O3 in response to increased NOx emissions are quite similar for both

mechanisms, although RADM2 shows slightly stronger NOx saturation (i.e., a stronger decrease in O3 given a 30% increase in NOx emissions) in the area centered around Benelux, and stronger NOx sensitivity over Scandinavia and northwest Russia."

7) lines 558-559: Mind also that the highest sensitivity for ozone production with regards to VOC is at the regions of high NOx emissions as someone would expect for the regions in the VOC limited regime.

We see that in areas with high NOx emissions such as Benelux, northern France and Germany, and shipping tracks in the Mediterranean, both RADM2 and MOZART predict VOC-sensitive conditions. This point have been added to the discussion in the revised manuscript. However, the increases in O3 with +30% VOC emissions are still relatively small. The text has been updated as follows:

"Areas where MOZART and RADM2 are in agreement in predicting VOC sensitivity (increased O3 concentrations in response to increased VOC emissions) are generally those with high NOx emissions, where one would expect the highest VOC sensitivity based on theory; these areas include Benelux, northern France, northwest Germany, and shipping tracks in the Mediterranean. However, the increase in O3 concentration is modest for both mechanisms; for RADM2 it is generally limited to increases of 2-4% over the base simulation."

8) lines 565-566: Do you think that the different O3 sensitivity to VOC changes in the two schemes can account for the O3 differences between RADM2 and MOZART (e.g. the lower ozone values in MOZART)? If yes, in which sense?

The results of the +30% VOC sensitivity studies for July (Figure 16) indicate that d[O3]/d[VOC] is higher (more positive) for RADM2 than for MOZART for the chemical regime represented by the models in July 2007. This is an indication that the two mechanisms are simulating different O3 chemical regimes – in the case of RADM2, there is a greater extent of VOC sensitivity, which means that addition of VOC emissions moves the chemistry in the direction of maximum O3 production efficiency, which

is not the case for MOZART over much of the domain. A more extensive study would be needed to evaluate whether the conclusion that d[O3]/d[VOC] is higher for RADM2 than for MOZART can be applied more generally. In our simulations, this effect (i.e., more O3 incremental production from VOC in RADM2 than in MOZART) appears to be dominated by other differences between the mechanisms (e.g., the inorganic rate coefficients), given that O3 concentrations predicted by MOZART are always greater than those predicted by RADM2 in our simulations. A discussion of this has been added to Section 4.3:

"The results of the +30% VOC sensitivity studies for July indicate that d[O3 ]/d[VOC] is higher (more positive) for RADM2 than for MOZART for the chemical regime represented by the models in July 2007. This shows that the two mechanisms are simulating different O3 chemical regimes – in the case of RADM2, there is greater VOC sensitivity, meaning that addition of VOC emissions moves the chemistry in the direction of maximum O3 production efficiency; this is not the case for MOZART over much of the domain. A more extensive study would be needed to evaluate whether the conclusion that d[O3 ]/d[VOC] is higher for RADM2 than for MOZART can be applied more generally."

9) lines 575-578: This is an interesting result which shows that differences in rate constants can account by 40% for the O3 differences between RADM2 and MOZART runs. You may highlight this result a bit more.

This result has been highlighted further in the Abstract and in the Summary and Conclusions. In the Summary and Conclusions section, we further suggest that harmonization of inorganic rate constants could potentially lead to reduced spread in predicted O3 among multi-model studies such as AQMEII. In the abstract, discussion of this difference now reads: "Additionally, we found that differences in reaction rate coefficients for inorganic gas phase chemistry in MOZART- 4 vs. RADM2 accounted for a difference of 8 $\mu$g m$-3$ , or 40% of the summertime difference in O3 predicted by the two mechanisms."

In the Summary and Conclusions, the text has been updated as follows. The first sentence was in the original manuscript, the second sentence has been added.

"Although the most fundamental differences between MOZART- 4 and RADM2 (and other chemical mechanisms used in regional modeling) is the representation of VOC oxidation chemistry, we find that approximately 40% of the difference seen in predicted O3 seen in this study can be explained by differences in inorganic reaction rate coefficients employed by MOZART- 4 and RADM2. This result suggests that harmonization of inorganic rate coefficients among chemical mechanisms used for regional air quality modeling might be valuable, and could potentially lead to a smaller spread in model-predicted O3 compared to that seen in, e.g., the multi-model studies of AQMEII (Solazzo et al., 2012b; Im et al., 2015)."

10) lines 591-594: Taking into consideration all three (rate constants, deposition and photolysis schemes) it seems that altogether account about 60% for the O3 differences between RADM2 and MOZART runs. Is this correct? You may highlight this conclusion.

It is true that if one looks at the average change in O3 concentration in these three sensitivity simulations, then a total of 60% of the MOZART-RADM2 difference in O3 concentration is explained, assuming that the effects are additive. However, the authors have consciously avoided presenting this as a conclusion in the text; since the effects of inorganic rate constants, photolysis and deposition are highly interconnected, it is reasonable to assume their combined effects may not be simply additive. We consider a quantification of the nonlinearity of this behavior to be outside the scope of this study.

11) Figure 3: I guess here the authors refer to wind direction. Please also provide information on the approach calculating the wind direction difference between obs and model.

The caption for Figure 3 has been fixed and now correctly refers to wind direction rather than wind speed. A more detailed description of how modeled wind direction was compared to observed wind direction has been added to Section 3.3, and reads

as follows.

"When applying these statistics to wind direction, wind direction was treated as a scalar quantity, when in fact it is a vector. This simple approach was favored rather than applying a correction (as done by, e.g., Zhang et al. (2013a) in cases where the difference in modeled vs. observed wind direction were greater than 180°). This is not expected to make an important impact on our analysis, especially since northerly winds (i.e., centered around 0°, or equivalently 360°) are not prevalent in Europe (see Figure 3 and Figure S2 in the Supplementary Material)."

12) Figure 16: Maybe it would be better to show the sensitivity result in a percentage scale (from -10 to 10 %).

In Figure 16, the plot has been adjusted to show the percent difference rather than the fractional difference.

Minor comments line 209: delete double "and". line 239: It is "for" instead of "fo". line 305: Maybe "related" instead of "associated" . line 406: It is "configuration" instead of " configuruation".

All of the above minor comments have been addressed with corrections in the text.

lines 427-429: The sentence needs rephrasing. It is not clear.

The sentence now reads "Coates et al. (2016) have shown that adding representation of stagnant conditions (which were not represented in Knote et al. (2015)) to a box model increased the sensitivity of predicted O3 to the chemical mechanism, and also improved model agreement with observations."

We believe this has improved the clarity of the original sentence, which read "Coates et al. (2016) have shown that accounting for stagnant conditions in a box model increased the variability in predicted O3 with temperature in a way that better reproduced the variability seen in observational datasets and 3-D model simulations; adding representation of stagnant conditions (which were not represented in Knote et al. (2015)) to the

box model also increased the sensitivity of predicted O3 to the chemical mechanism."

[Figure]

**Fig. 1.** Figure S2, Seasonally-Averaged Wind Vectors from ERA-Interim

[Figure]

**Fig. 2.** Figure S12, CH2O/NOy indicator for July sensitivity run